**RESEARCH**                                                                    **Open Access**

# Systematic functional interrogation of human pseudogenes using CRISPRi

Ming Sun[1,2†], Yunfei Wang[1,3†], Caishang Zheng[1†], Yanjun Wei[1], Jiakai Hou[1,4], Peng Zhang[1,5], Wei He[6,7,8], Xiangdong Lv[9,10,11], Yao Ding[9,10,11], Han Liang[1,8,12], Chung-Chau Hon[13], Xi Chen[9,10,11], Han Xu[1,6,7,8*] and Yiwen Chen[1,8*]

* Correspondence: Hxu4@ mdanderson.org; ychen26@ mdanderson.org
†Ming Sun, Yunfei Wang and Caishang Zheng contributed equally to this work.
[1]Department of Bioinformatics and Computational Biology, The University of Texas MD Anderson Cancer Center, Houston, TX 77030, USA
Full list of author information is available at the end of the article

## Abstract

**Background:** The human genome encodes over 14,000 pseudogenes that are evolutionary relics of protein-coding genes and commonly considered as nonfunctional. Emerging evidence suggests that some pseudogenes may exert important functions. However, to what extent human pseudogenes are functionally relevant remains unclear. There has been no large-scale characterization of pseudogene function because of technical challenges, including high sequence similarity between pseudogene and parent genes, and poor annotation of transcription start sites.

**Results:** To overcome these technical obstacles, we develop an integrated computational pipeline to design the first genome-wide library of CRISPR interference (CRISPRi) single-guide RNAs (sgRNAs) that target human pseudogene promoter-proximal regions. We perform the first pseudogene-focused CRISPRi screen in luminal A breast cancer cells and reveal approximately 70 pseudogenes that affect breast cancer cell fitness. Among the top hits, we identify a cancer-testis unitary pseudogene, MGAT4EP, that is predominantly localized in the nucleus and interacts with FOXA1, a key regulator in luminal A breast cancer. By enhancing the promoter binding of FOXA1, MGAT4EP upregulates the expression of oncogenic transcription factor FOXM1. Integrative analyses of multi-omic data from the Cancer Genome Atlas (TCGA) reveal many unitary pseudogenes whose expressions are significantly dysregulated and/or associated with overall/relapse-free survival of patients in diverse cancer types.

**Conclusions:** Our study represents the first large-scale study characterizing pseudogene function. Our findings suggest the importance of nuclear function of unitary pseudogenes and underscore their underappreciated roles in human diseases. The functional genomic resources developed here will greatly facilitate the study of human pseudogene function.

**Keywords:** Pseudogene, Unitary pseudogene, CRISPR interference, Cancer, Luminal A breast cancer, Nucleus, FOXA1, Transcriptional regulation, FOXM1, TCGA, GTEx

## Introduction

Pseudogenes are defined as dysfunctional copies of protein-coding genes that have lost their coding potential due to the accumulation of disruptive mutations such as premature stop codons and frame-shift insertions/deletions [1, 2]. The corresponding protein-coding paralogs of pseudogenes are referred to as parent genes. Pseudogenes are evolutionary relics present in the genomes of a wide variety of species, including bacteria, plant, and metazoans [3, 4]. They are often lineage-specific throughout the evolution, and mammalian genomes contain much more pseudogenes than other metazoan species [4]. Based on their generation mechanism during the course of evolution, pseudogenes can be categorized into three major classes: (1) unprocessed (also referred to as duplicated) pseudogenes, derived from duplication of protein-coding genes; (2) processed pseudogenes, generated by retrotransposition of mRNA transcribed from protein-coding genes back into the genome; and (3) unitary pseudogenes, which arise through mutations in previously functional protein-coding genes without gene duplication. Unitary pseudogenes have no functional protein-coding gene counterparts in the same genome, but only have functional coding orthologs in the genome of other organisms [5]. Different species are often enriched for different classes of pseudogenes. For example, in metazoans, the mammalian genomes are dominant by processed pseudogenes [4], while worm, fly, and zebrafish genomes are enriched for unprocessed pseudogenes [4].

Pseudogenes have long been considered as nonfunctional genomic elements. However, recent studies revealed multiple examples of pseudogenes that can exert important regulatory function at the RNA level [6–10], through two major mechanisms. One mechanism is via a production of small RNAs such as small interference RNA (siRNA) [9, 10], whereby pseudogene-derived small RNAs can exert regulatory functions. The other mechanism is via sponge RNA [11] or competing endogenous RNA (ceRNA) [12] regulation, in which pseudogenes cross-regulate the expression of their parent genes or other protein-coding genes by competing for binding of the same set of microRNAs (miRNAs) [6–8]. The ceRNA regulation remains controversial as a general mechanism of gene regulation under physiological conditions [13–15], because growing evidence suggests that many active miRNAs are probably not susceptible to ceRNA competition and ceRNA regulation may be highly context-specific [14, 15]. Many pseudogenes showed dysregulated [16] or subtype-specific expression [17] in cancer, suggesting their important role in the etiology of complex diseases.

Despite an increasing appreciation of the functional significance of pseudogenes, it remains a technical challenge to interrogate their functions at a large-scale because pseudogenes usually share high sequence similarity with their parent genes, except for unitary pseudogenes. This high sequence homology makes it difficult to use traditional loss-of-function approaches such as RNA interference (RNAi) or locked nucleic acid (LNA) to target pseudogene transcripts while leaving the transcripts of their parent genes intact. To date, there has been no systematic large-scale effort to characterize pseudogene function. As a result, function of the vast majority pseudogenes remains unknown, except for only a handful of cases.

To fill this gap, we leveraged the recently emerged genetic perturbation technique, CRISPR (clustered regularly interspaced short palindromic repeat) interference (CRISPRi) [18–20] and developed an integrated computational pipeline to design CRISPRi

single-guide RNAs (sgRNAs) that target pseudogene promoters, which are often readily distinguishable from that of their parent genes, allowing for a systematic interrogation of pseudogene function in human cells. Different from the gene knockout with wild-type CRISPR/Cas9, CRISPRi uses a catalytically inactive Cas9 (dCas9) fused with a transcriptional repressor, targeted through sgRNAs, to specific genomic loci in the promoter-proximal regions to repress gene expression [19]. The majority of human pseudogenes are processed pseudogenes. The promoter sequences of processed pseudogenes and their parent genes are generally more divergent than their transcript sequences because processed pseudogenes usually lack 5′ promoter sequence of their parent genes. Compared with the RNAi/LNA-based method, CRISPRi offers a unique advantage to specifically inhibit the expression of pseudogenes, without directly interfering parent gene transcription. In addition, unlike RNAi that tends to be less effective for silencing nuclear RNAs, CRISPRi modulates gene expression at the transcriptional level and can thus suppress pseudogene expression regardless of RNA sub-cellular localization. Leveraging this CRISPRi sgRNA library specially designed for human pseudogenes, we performed to date the first pseudogene-focused pooled CRISPRi screen in luminal A breast cancer cells to systematically identify human pseudogenes that are critical for breast cancer cell fitness.

## Results

### An integrated computational pipeline for designing CRISPRi sgRNAs that target human pseudogenes

We developed an integrated computational pipeline to design a CRISPRi sgRNA library for the annotated human pseudogenes. As the effectiveness of CRISPRi-based transcriptional repression relies heavily on the precise recruitment of the effector complex to the target gene transcription start site (TSS) [19], an accurate annotation of the TSSs of individual genes is essential to the success of designing effective sgRNAs. However, the TSSs of pseudogenes are relatively poorly annotated in comparison with that of protein-coding genes. To address this issue, we first integrated the FANTOM5 cap analysis of gene expression (CAGE) [21] data with GENCODE V22 transcriptome annotation to define the TSSs on a transcriptome-wide level, including the pseudogenes (Fig. 1A and "Methods"), as described previously [22]. A total of 97,074 CAGE clusters were assigned as the TSSs of transcripts in GENCODE V22. Next, we used the algorithm Sequence Scan for CRISPR (SSC) [23] to scan for sgRNA targets based on the genomic sequence within a 500-bp window centered on each TSS. After filtering out sgRNA sequences of low quality from the initial 770,965 sgRNAs, we selected 359,082 uniquely mapped sgRNAs that target the TSS-proximal regions of 42,609 genes. There was a total of 57,031 uniquely mapped sgRNAs that target TSS-proximal regions of 7762 pseudogenes (Additional File 1: Table S1).

As a proof-of-principle systematic study of human pseudogene function with the designed CRISPRi sgRNAs, we focused on breast cancer, which is the most commonly diagnosed cancer and the leading cause of cancer death in women worldwide and has well-defined molecular subtypes [24]. Luminal A is the most common subtype and triple negative/basal-like is a more aggressive subtype in breast cancer. To generate a CRISPRi sgRNA library that target the expressed pseudogenes in MCF7 and MDA-

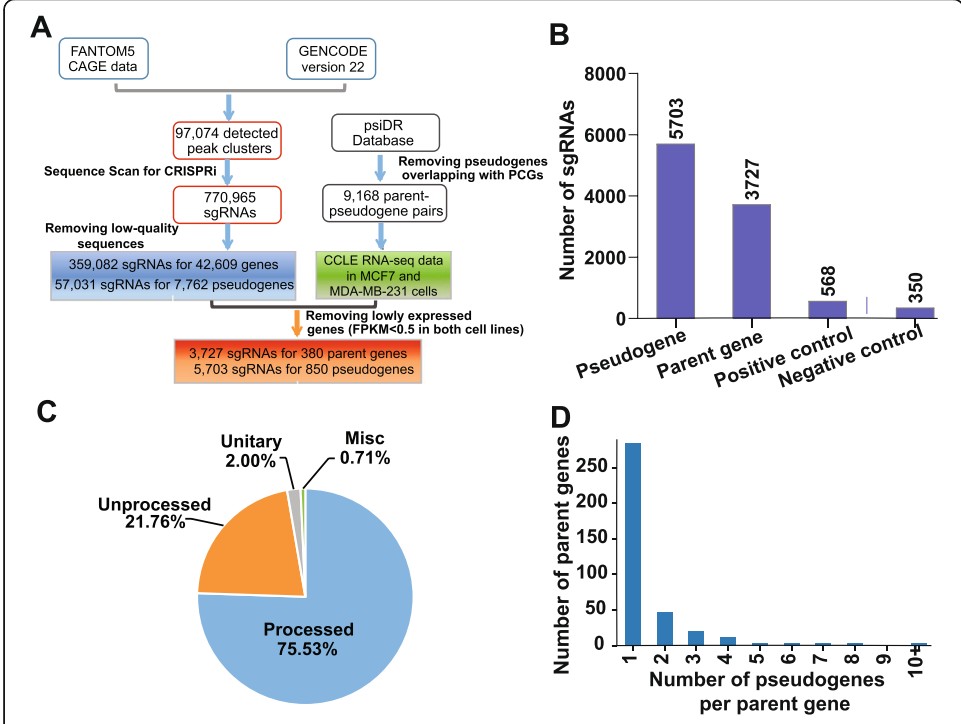

**Fig. 1** An integrated computational pipeline for designing CRISPRi sgRNA library to screen for functional human pseudogenes and parent genes. **A** A workflow of the sgRNA library design for pseudogene-focused CRISPRi screen. **B** The number of sgRNAs targeting pseudogene, parent genes, and positive and negative control sgRNAs that were included in the screen. **C** The pie chart showing the percentage of different types of pseuodgenes included in the screen. **D** The distribution of the number of pseudogenes per parent gene that were included in the screen

MB-231 cell lines, the two breast cancer cell line models representing luminal A and triple negative/basal-like breast cancer, we used the RNA-seq data from The Cancer Cell Line Encyclopedia (CCLE) [25] to filter out the lowly expressed pseudogenes (FPKM < 0.5 in both breast cancer cell lines). We also filtered out the pseudogenes with less than three designed sgRNAs. After filtering, there were 5703 designed sgRNAs corresponding to 850 pseudogenes, with the median number of 6 sgRNAs per pseudogene (Additional File 1: Table S1; Additional File 2: Fig. S1A). Furthermore, to interrogate the function of both parent genes (if available) and pseudogenes in the same screen, sgRNAs targeting high-confidence parent genes [26] (Fig. 1A) that passed the same filters as for pseudogenes were included in the library, resulting in 3727 sgRNAs that target 380 parent genes ("Methods," Additional File 1: Table S1). In addition to 9430 gene-specific CRISPRi sgRNAs targeting pseudogenes and parent genes, we included 568 sgRNAs targeting 71 core fitness genes as positive controls, and 267 sgRNAs targeting AAVS1 and 83 non-targeting sgRNAs as negative controls, as described previously [27] (Fig. 1B, Additional File 1: Table S1). The pseudogenes in our screen covered all three major classes of pseudogenes, including processed (*n* = 642), unprocessed (*n* = 185), and unitary pseudogenes (*n* = 17), as well as non-classified/misc (*n* = 6) pseudogenes (Fig. 1C). For the majority of parent genes that were included, there were no more than two corresponding pseudogenes with targeting sgRNAs in the library (Fig. 1D).

### A CRISPRi screen identifies functional human pseudogenes of different categories

To identify the pseudogenes that critically contribute to luminal A breast cancer cell growth and/or survival (fitness), we conducted a pooled CRISPRi screen (Fig. 2A and "Methods"). We conducted the screen in MCF7 cell line that stably express the streptococcus pyogenes (Sp) dCas9-KRAB fusion protein [20] in triplicates, in a similar way to the CRISPR-Cas9 or CRISPRi screen performed previously [20, 28]. Briefly, the oligonucleotides containing both sgRNAs and flanking linker sequences were synthesized as a pooled library and the resultant library was amplified and cloned into the lentiviral vectors. Cells transduced with the lentiviral vectors encoding sgRNA library were selected with puromycin (puro). The puro-selected cells were then split into three replicates and passaged for 21 days. We collected individual replicates on day 0 (D0) and day 21 (D21). The abundance change of individual sgRNAs between the initial and final cell populations were quantified by next-generation sequencing to identify the genes that are important for cell fitness. The sgRNA abundance from the three replicates showed a significant correlation ($p < 2.2 \times 10^{-16}$) with each other on day 0 and day 21 (Additional File 2: Fig. S1B). As expected for the working positive controls, we observed a notable depletion in the abundance of the sgRNAs targeting positive control core

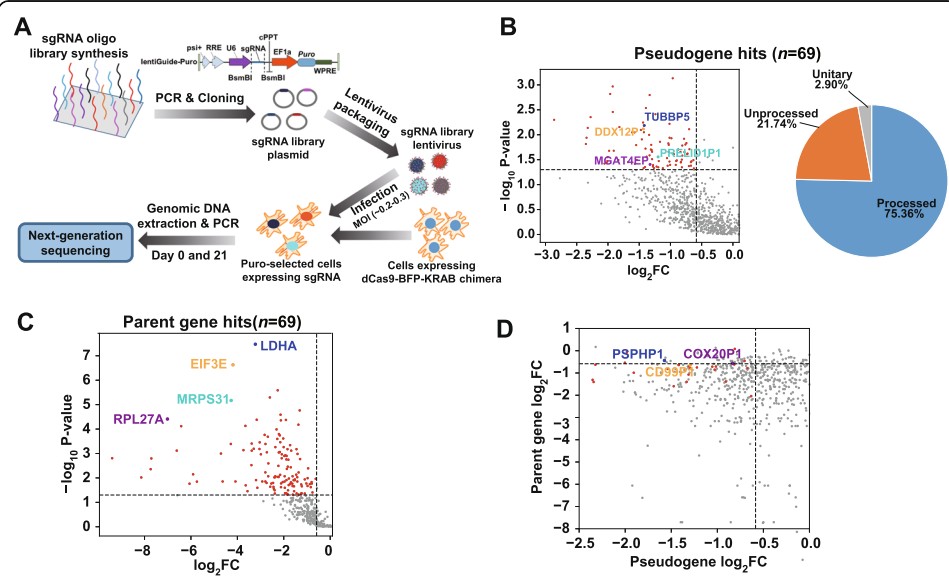

**Fig. 2** CRISPRi screen reveals functional human pseudogenes. **A** Schema depicting the workflow for construction of lentiviral vectors encoding sgRNA library and experimental design of CRISPRi screens. **B** The scatter plot showing the $\log_2$ Fold-Change (FC) and the statistical significance ($-\log_{10}P$ value) of sgRNA abundance difference between day 21 and day 0 for the negatively selected pseudogenes ($\log_2$ FC < 0), and the pie chart showing the percentage of unitary, processed, and unprocessed pseudogenes among all pseudogenes hits in MCF7 cells. The dots corresponding to the pseudogene hits are shown in red. The examples of pseudogene hits MGAT4EP, DDX12P, TUBBP5, and PRELID1P1 are highlighted in different colors. **C** The scatter plot showing the $\log_2$FC and the statistical significance ($-\log_{10}P$ value) of sgRNA abundance difference between day 21 and day 0 for the negatively selected parent genes ($\log_2$FC < 0). The dots corresponding to the parent gene hits are shown in red. The examples of parent gene hits RPL27A, MRPS31, EIF3E, and LHDA are shown in different colors. **D** The scatter plot showing the $\log_2$FC of sgRNA abundance difference between day 21 and day 0 for pseudogenes and their corresponding parent genes. The dots corresponding to the pseudogene hits, whose parent gene does not pass the statistical significance threshold, are shown in red. The examples of these pseudogene hits PSPHP1, COX20P1, and CD99P1 are shown in different colors

essential genes (Additional File 2: Fig. S1C and D) in final cell populations (D21) compared with the initial ones (D0). Interestingly, the sgRNAs targeting parent genes also showed a general decrease in abundance in final cell populations (Additional File 2: Fig. S1C and D), suggesting that many of them are essential to cell fitness. Moreover, we used the parent genes included in our library that were previously identified as essential genes in MCF7 cells by CRISPR-Cas9 knockout screens [29] to serve as independent positive controls. As expected, the sgRNAs that target the essential parent genes previously identified by CRISPR-Cas9 knockout screens showed a statistically significant larger fold change of decrease between D21 and D0, compared with the ones targeting all parent genes (Mann-Whitney $U$ test, $p < 1.21 \times 10^{-4}$, Additional File 2: Fig. S1D). Similar result was observed in the gene-level histograms where the fold change of each gene was represented by the 2nd largest fold change of its corresponding sgRNAs, based on the output of MAGeCK program [30] (Additional File 2: Fig. S1E).

To identify the genes that are essential to cell fitness, we used MAGeCK [30] to assess the statistical significance of the level of sgRNA depletion and identify the genes that were under significantly negative selection in the screens ($p < 0.05$, FDR < 0.25 and $\log_2$Fold-Change≤-$\log_2(1.5)$, "Methods"). Bidirectional promoters are an important source of off-target effect for CRISPRi and can result in false positives in cell fitness screens [27]. To control for the false positives caused by bidirectional promoters, we excluded the negatively selected genes from screen hits, if their TSSs were within 1 kb from the TSSs of another gene based on GENCODE V22 annotation ("Methods," Additional File 3: Table S2). We used 1 kb as a cutoff, based on our previous finding that genes located up to 1 kb from an essential gene are more likely to be scored as an essential one in a fitness screen, due to CRISPRi off-target effect [27]. After filtering out the potential false positives due to bidirectional promoters, we identified 69 pseudogene hits (out of 850 pseudogenes) that were negatively selected in MCF7 cells (Fig. 2B). In contrast, we did not find any significant positively selected genes in the screen ($p <$ 0.05, FDR < 0.25 and $\log_2$Fold-Change≥$\log_2(1.5)$, Additional File 3: Table S2). The negatively selected pseudogene hits contained all three major classes of pseudogenes (Fig. 2B). In addition, we identified 69 parent gene hits (out of 380 parent genes) that were negatively selected (Fig. 2C). Interestingly, we found that the negatively selected parent genes showed a larger magnitude of sgRNA depletion than their corresponding pseudogenes (paired *t*-test, $p < 3.03 \times 10^{-52}$, Fig. 2D), suggesting that in general, parent genes are functionally more important for cell fitness than their pseudogene counterpart. However, a small fraction of pseudogenes showed a larger magnitude of sgRNA depletion than their corresponding parent genes, suggesting that the function of these pseudogenes might be less dependent on their parent genes (Fig. 2D).

To investigate the effect of potential off-targeting sgRNAs on the screen results, we used Cas-OFFinder [31] to predict the putative off-target sites of individual sgRNAs in the human genome ("Methods"). Because the off-target effect is much weaker when there are > 1 nucleotides (nt) of mismatches [32] or there is any RNA/DNA bulge [33] in the potential off-target sites, we focused on the predicted off-target sites with 1-nt mismatch from a given sgRNA sequence. We found that most sgRNAs targeting pseudogene/parent gene were associated with no or very small number (≤ 1) of predicted off-target sites in the human genome (Additional File 2: Fig. S1F). Moreover, the number of predicted genomic off-target sites associated with off-targeting sgRNAs did

not show significant difference between pseudogene/parent gene hits and non-hits (Mann-Whitney $U$ test, $p \geq 0.18$, Additional File 2: Fig. S1G). Importantly, we found that the pseudogene/parent gene hits did not have a significantly larger proportion of off-targeting sgRNAs with a large number ($\geq 10$) of off-target sites, compared with the other pseudogenes/parent genes (Fisher's exact test, $p > 0.32$). Collectively, these results suggest that in overall, the potential off-targeting sgRNAs may have little impact on differentiating the screen hits from the other genes and thus the results of our CRISPRi screens. Aside from the global analysis of potential off-target effect, we further investigated whether the pseudogene hits identified from our screen could be confounded by the potential off-targeting sgRNAs from its corresponding parent genes or vice versa. We found that out of the 69 pseudogene and 69 parent gene hits, 15 pseudogenes and their corresponding parent genes were both identified as hits. Among a total of 30 (15 pseudogene and 15 parent gene) hits, we found 6 of them have one and the only one significant negatively selected sgRNA ($p < 0.05$, log$_2$Fold-Change$\leq$-log$_2$(1.5)) that harbors a predicted off-target site within [$- 2$ kb,$+ 1$ kb] from the TSS of its corresponding pseudogene/parent gene ("Methods," Additional File 3: Table S2). After removing the only one putative functional off-targeting sgRNA for all six pseudogene/parent gene hits, four of them still had at least two significant negatively selected sgRNAs and two of them had one significant negatively selected sgRNA. These results indicate that the vast majority of the pseudogene/parent gene hits are not confounded by the predicted off-targeting sgRNAs from their corresponding pseudogenes/parent genes.

### Validating top pseudogene hits with an upregulated expression in breast cancer

To validate the top pseudogene hits from our screen that are relevant to breast cancer, we focused on the pseudogenes, whose targeting sgRNAs showed the strongest growth inhibitory effect in MCF7 cells and that were significantly upregulated in breast cancer (Fig. 3A), compared with normal breast tissues (log$_2$Fold-Change$\geq$log$_2$(1.5) and FDR < 0.05, "Methods"). We selected four candidates DDX12P, TUBBP5, MGAT4EP, and PRELID1P1 that had at least 50% effective and negatively selected sgRNAs scored by MAGeCK for functional validation. To determine the role of these four pseudogenes in cancer cell fitness, we examined their loss-of-function phenotype in MCF7 cells with CRISPRi-mediated gene silencing. For each pseudogene, we selected the top two sgRNAs (> 55 bp apart from each other in the genome) that showed the strongest growth inhibitory effect in CRISPRi screen for gene silencing ("Methods"). Real-time quantitative reverse transcription PCR (qRT-PCR) experiments confirmed that the selected gene-specific sgRNAs effectively reduced RNA level of the corresponding pseudogenes in comparison with the non-targeting sgRNA (Fig. 3B, Additional File 4: Table S3). The effective depletion of these pseudogenes by either of the two gene-specific sgRNAs inhibited the growth of MCF7 cells (Fig. 3C–F) and impaired their clonogenic capacity (Fig. 3G, H). To assess the robustness of our observations to the use of different negative controls, we included two genome-targeting negative control sgRNAs with one targeting the Adeno-Associated Virus Integration Site 1 (AAVS1) region (sg-AAVS1) and the other targeting the genomic region distant from AAVS (sg-nAAVS1) in the loss-of-function study (Additional File 4: Table S3). The sg-AAVS1 and sg-nAAVS1 were selected from our CRISPRi sgRNA library and a CRISPR-Cas9 sgRNA

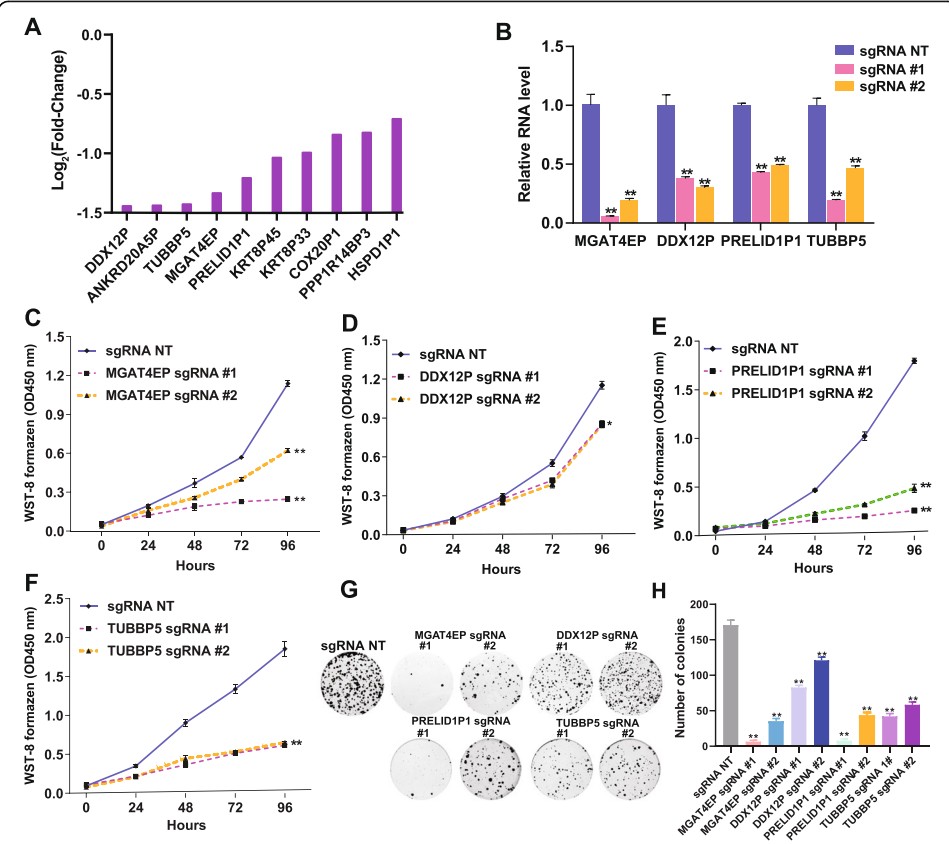

**Fig. 3** Validation of top pseudogene hits in MCF7 cells. **A** A bar graph shows the log$_2$FC of sgRNA abundance difference between day 21 and day 0 for the top-ranked (by log$_2$FC) pseudogene hits in MCF7 cells that showed a significant upregulation in breast cancer compared with normal breast tissues based on TCGA data. **B** qRT-PCR analysis of the RNA level of MGAT4EP, DDX12P, PRELID1P1, and TUBBP5 in MCF7-dCas9 cells transduced with negative control non-targeting sgRNA (sg-NT) or gene-specific sgRNA. GAPDH was used as an internal control. The growth of MCF7-dCas9 cells transduced with sg-NT or gene-specific sgRNA for **C** MGAT4EP, **D** DDX12P, **E** PRELID1P1, and **F** TUBBP5 was monitored (OD450 absorbance for WST-8 formazen) every 24 h with CCK-8 assay for 96 h. **G** The representative pictures of clonogenic growth and **H** the bar graph quantifying the colonies formed by MCF7-dCas9 cells transduced with sg-NT or gene-specific sgRNAs for MGAT4EP, DDX12P, PRELID1P1, and TUBBP5, after cells were cultured for 2 weeks. All data are shown as mean ± standard deviation (SD), $n = 3$. The Student's $t$ test was used to assess the statistical significance of difference in mean between two experimental groups (*$p < 0.05$; **$p < 0.01$; ns: not significant, $p \geq 0.05$)

library used in previous knockout screens [34], respectively, because they showed an insignificant abundance change across conditions. Similar to the case of using a non-targeting sgRNA as a negative control, gene-specific sgRNAs effectively reduced RNA level of the corresponding pseudogenes compared with negative controls of genome-targeting sgRNAs (Additional File 2: Fig. S2A). The effective depletion of these pseudogenes consistently impaired the clonogenic capacity of MCF7 cells and inhibited their growth (Additional File 2: Fig. S2B and C) when the genome-targeting sgRNAs were used as a negative control. Taken together, these results indicate that the loss-of-function phenotypes we observed are robust to different negative controls. To rule out the possibility that the observed loss-of-function phenotype for these pseudogenes are caused by CRISPRi-mediated off-target effect, we further performed rescue experiments, by overexpressing the corresponding cDNAs of these pseudogenes in the

presence of CRISPRi-mediated knockdown. We found that cDNA expression was able to rescue the CRISPRi-mediated loss-of-function phenotype in both cell growth and clonogenic formation (Additional File 2: Fig. S2D and E). These results indicate that the observed loss-of-function phenotypes for these four pseudogenes are not due to off-target effect. In addition, we aligned the spacer sequences of the sgRNAs used in the validation experiments against the promoter sequences of the corresponding parent genes (if available) and chose the first PRELID1P1-targeting sgRNA (Additional File 4: Table S3), which only has 1-nt mismatch to the promoter of its parent gene, to assess its specificity in inhibiting pseudogene transcription. Importantly, we found CRISPRi-mediated silencing by this sgRNA significantly reduced Pol II binding to the promoter of PRELID1P1, but not to that of its parent gene PRELID1 (Additional File 2: Fig. S3A). This result confirmed that our CRISPRi screen enabled transcriptionally inhibiting pseudogene transcription without directly interfering with the transcription of its parent gene, which is critical for systematically interrogating the function of pseudogenes independent from their parent genes.

### MGAT4EP is a cancer-testis unitary pseudogene that promotes the growth of breast cancer cells

Among the four pseudogene hits that we validated, MGAT4EP showed a consistently strong loss-of-function phenotype in both cell growth and clonogenic assay, and a larger fold change in expression between breast cancer and normal breast tissues. Therefore, we focused on this unitary pseudogene for a detailed functional characterization. Interestingly, MGAT4EP not only showed a significant upregulation in breast cancer compared with normal breast tissue based on the Cancer Genome Atlas [35] (TCGA) RNA-seq data [36], but also showed a much higher expression in testis than other normal tissues based on the RNA-seq data generated by the Genotype-Tissue Expression (GTEx) project [37] (Additional File 2: Fig. S3B and C), indicating that MGAT4EP is a cancer-testis unitary pseudogene. We further performed 5′ and 3′ RACE (rapid amplification of 5′/3′ complementary DNA ends), and confirmed that the experimentally determined 5′ and 3′ end of MGAT4EP transcript (NR_038135.2) were consistent with its original RefSeq annotation (Additional File 2: Fig. S3D).

Some of the human pseudogenes were found to undergo translation and might express functional proteins [38]. To rule out the possibility that MGAT4EP encodes a protein/micropeptide and has a coding-dependent function, we first analyzed the publically available [39] and in-house ribosome profiling (ribo-seq) data (unpublished) in MCF7 cell line ("Methods") and found no ribo-seq reads that support the ribosome occupancy on MGAT4EP RNAs. Second, we predicted putative ORFs encoded by MGAT4EP using an ORF prediction module that solely relies on the sequence information and is implemented in the Ribo-TISH package [40] ("Methods"), and searched the publically available mass-spectrometry (MS) data in MCF7 and T47D cells [41] for the MS/MS spectra that matched the protein sequences corresponding to these putative ORFs ("Methods"). We found no MS evidence of the protein products encoded by these putative ORFs. Finally, we performed an in vitro translation assay ("Methods") and found no evidence of any protein products generated by MGAT4EP translation (Additional File 2: Fig. S3E). These results indicate that MGAT4EP does not encode a protein/micropeptide and it functions as an ncRNA.

To validate the function of MGAT4EP using an alternative loss-of-function approach to CRISPRi, we performed siRNA-mediated silencing of MGAT4EP to assess its effect on cell growth ("Methods"). Consistent with the results obtained by CRISPRi method, we found that the effective siRNA-mediated depletion of MGAT4EP (Additional File 2: Fig. S3F) inhibited the growth of both MCF7 and T47D, the two independent luminal A breast cancer cell lines (Additional File 2: Fig. S3G).

### MGAT4EP is predominantly localized in the nucleus and interacts with transcription factor FOXA1

An important mechanism, whereby many pseudogenes [6–8] exert their function, is through competing for miRNA binding with its parent gene or other protein-coding genes, the so-called sponge [11]/ceRNA [12, 42] mechanism. This mechanism is not specific to pseudogenes that have parent genes with high sequence homology (i.e., processed/unprocessed pseudogenes). A previous study [8] revealed that *Pbcas4*, a mouse unitary pseudogene that does not have functional protein-coding counterparts in the mouse genome and lost its protein-coding capability specifically during rodent evolution, can also serve as a ceRNA with conserved miRNA target sites. A critical factor determining the efficacy of a sponge/ceRNA regulation is the cytoplasmic localization of the involved RNAs [43] where most miRNA-based regulation occurs. Because sub-cellular localization is important for dictating mechanism of pseudogene function, we determined sub-cellular localization of MGAT4EP, using nuclear/cytoplasmic fractionation coupled with qRT-PCR. The good quality of the nuclear/cytoplasmic fractionation was supported by the enrichment of nuclear/cytoplasmic protein (Additional File 2: Fig. S3H) and RNA controls (Fig. 4A) in their respective sub-cellular compartments. Interestingly, we found that MGAT4EP was dominantly localized in the nucleus (Fig. 4A), suggesting that it may exert a nuclear function, which is distinct from a traditional sponge/ceRNA mechanism.

To infer its nuclear function and systematically identify the nuclear proteins that may form physical interaction with MGAT4EP, we used an RNA pull-down method [44] ("Methods") based on in vitro transcription of 3′ end-biotin-labeled RNAs and affinity purification of RNA-interacting nuclear proteins, followed by mass spectrometry (MS). The antisense (AS) sequence of MAGAT4EP transcript (NR_038135.2) was used as a negative control RNA in pull-down experiment, to filter out non-specific interactions (Fig. 4B). We found that among the proteins specifically identified in MAGAT4EP pull-down group (i.e., at least two unique MS-identified peptides in the MAGAT4EP group and zero MS-identified peptide in the AS negative control group), eight transcription factors/epigenetic regulators showed significant upregulation in the luminal A breast cancer subtype compared with normal breast tissues (Additional File 2: Fig. S4A). Notably, one of them is FOXA1 (forkhead box A1), also known as HNF3α (hepatocyte nuclear factor 3α), a transcription factor that impacts estrogen receptor signaling and is key to mammary ductal development and progression of luminal A subtype breast cancer [45]. We confirmed by western blot analysis that FOXA1 was enriched by MAGAT4EP RNA pull-down (Fig. 4C). In contrast, SP1, a negative control protein that was not identified by MS, was not detected (Fig. 4C). To identify the regions in the MGAT4EP RNA that was required for its interaction with FOXA1, we generated four serial deletion mutants with the deletion of 1–700, 700–1400, 1400–2100, or 2100–2819 bps,

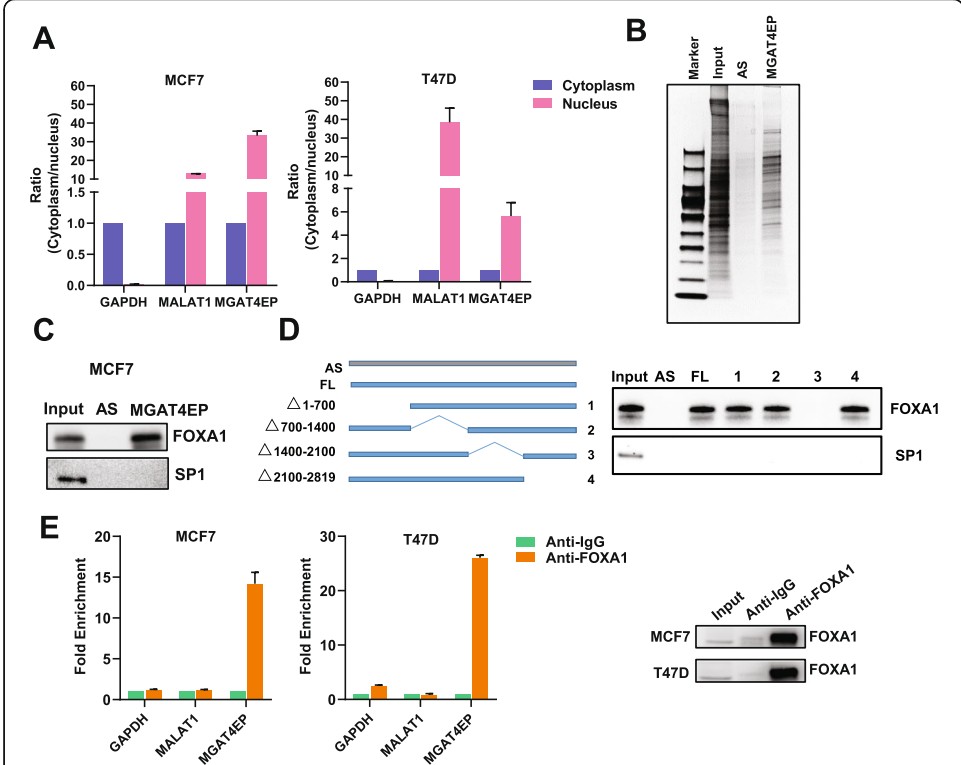

**Fig. 4** MGAT4EP is predominantly localized in the nucleus and interacts with transcription factor FOXA1. **A** The RNA level of MGAT4EP in nuclear and the cytoplasmic fraction of MCF7 and T47D cells was measured by qRT-PCR. MALAT1 RNA and GAPDH mRNA was used a positive control for nuclear and cytoplasmic fraction, respectively. **B** The proteins retrieved by RNA pull-down with MGAT4EP RNA and negative control antisense RNA (AS) were visualized by silver staining and subject to mass spectrometry (MS) analysis. **C** RNA pull-down coupled with western blot validated the interaction between MGAT4EP and FOXA1 that was identified from MS analysis. SP1 that was not found in MS analysis was used as a negative control. **D** RNA pull-down of the antisense, full-length, and serial deletion mutants of MGAT4EP RNA followed by anti-FOXA1/anti-SP1 western blotting. The four serial deletion mutants of MGAT4EP RNA were generated by deleting 1–700, 700–1400, 1400–2100, or 2100–2819 bps, respectively. **E** RIP-qPCR analysis with anti-FOXA1 or anti-IgG antibody validated the association of FOXA1 with MGAT4EP RNA, where MALAT1 and GAPDH RNA were used as negative controls. All data are shown as mean ± SD, $n = 3$. The Student's *t* test was used to assess the statistical significance of difference in mean between two experimental groups (*$p < 0.05$; **$p < 0.01$; ns: not significant, $p \geq 0.05$)

respectively. The RNA pull-down of antisense, full-length, and serial deletion mutants of MGAT4EP RNA followed by anti-FOXA1 western blotting showed that the deletion of 1400–2100 bps of MGAT4EP abolished its interaction with FOXA1 (Fig. 4D), suggesting that this region is critical for MGAT4EP-FOXA1 interaction. We further performed RNA immunoprecipitation (RIP) coupled with qRT-PCR for MGAT4EP RNA and two negative controls: GAPDH mRNA for cytoplasmic RNAs and MALAT1 for nuclear RNAs, respectively. Indeed, FOXA1 was associated with MGAT4EP RNA, but not the negative control of GAPDH mRNA and MALAT1 (Fig. 4E).

## MGAT4EP upregulates the expression of FOXM1, a direct target of FOXA1, by enhancing FOXA1 binding to its promoter

To identify common protein-coding gene targets that are co-regulated by MGAT4EP and FOXA1 and are important for mediating their tumor-promoting function in

luminal A breast cancer cells, we performed an integrated analysis of the RNA-seq data generated from cells with/without sgRNA-mediated MGAT4EP depletion, FOXA1 ChIP-seq data in luminal A breast cancer cell lines [46], and TCGA breast cancer data [36] (Fig. 5A). First, using our RNA-seq data, we identified 103 downregulated protein-coding genes ($\log_2$Fold-Change$\leq -\log_2(1.5)$ and FDR < 0.05) in MGAT4EP knockdown

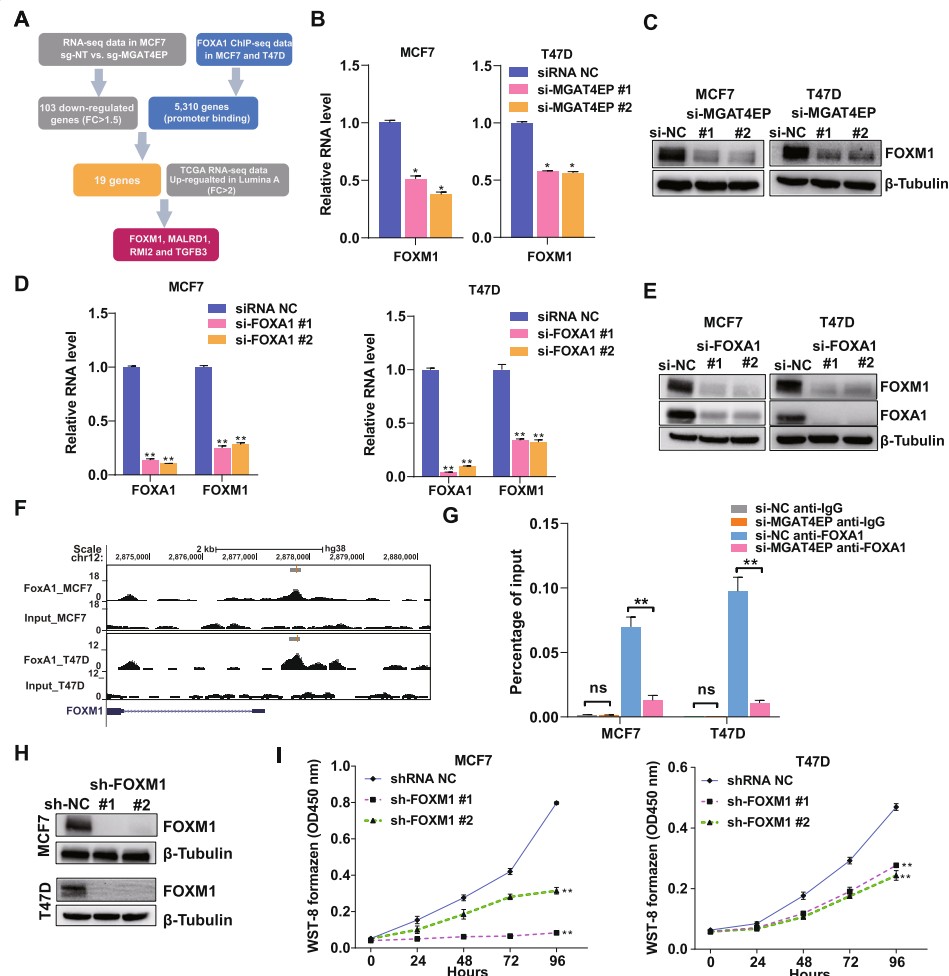

**Fig. 5** MGAT4EP upregulates the expression of FOXM1, a FOXA1 target, and enhances FOXA1 binding to its promoter. **A** Schema depicting the workflow of identifying potential protein-coding gene targets that were co-regulated by MGAT4EP and FOXA1 and were important for mediating their tumor-promoting function in luminal A breast cancer. **B** QRT-PCR analysis of FOXM1 mRNA expression and **C** western blot for measuring FOXM1 protein expression in MCF7 and T47D cells that were treated with negative control non-targeting siRNA (si-NC) or MGAT4EP-targeting siRNAs. **D** qRT-PCR analysis of FOXM1 mRNA expression and **E** western blot for measuring FOXM1 protein expression in MCF7 cells and T47D cells that were treated with si-NC or FOXA1-targeting siRNAs. **F** The signal track of FOXA1 ChIP-seq and the corresponding input in MCF7 and T47D cells. The identified ChIP-seq peaks were drawn as horizontal lines above the signal track. **G** ChIP-qPCR analysis was performed with anti-FOXA1 or anti-IgG antibody in MCF7 and T47D cells to confirm the enrichment of DNA fragments covering the FOXA1 ChIP-seq peak in the FOXM1 promoter. The effect of si-NC or MGAT4EP-targeting siRNAs on the binding of FOXA1 to the same region was assessed by ChIP-qPCR. **H** Western blot for measuring FOXM1 protein expression in MCF7 cells and T47D cells that were transduced with negative control non-targeting shRNA (sh-NC) or FOXM1-targeting shRNAs. **I** The growth of MCF7 and T47D cells transduced with sh-NC or FOXM1-targeting shRNAs was monitored (OD450 absorbance for WST-8 formazen) every 24 h with CCK-8 assay for 96 h. All data are shown as mean ± SD, $n = 3$. The Student's *t* test was used to assess the statistical significance of difference in mean between two experimental groups (*$p < 0.05$; **$p < 0.01$; ns: not significant, $p \geq 0.05$)

cells (Additional File 5: Table S4; Additional File 2: Fig. S4B). Next, using publicly available FOXA1 ChIP-seq data in MCF7 and T47D cells [46], we identified 5310 protein-coding genes that harbored at least one FOXA1 binding site in their promoter-proximal regions (− 1.5 kb, 0.5 kb) in either cell line. In total, 19 protein-coding genes showed downregulation upon MGAT4EP knockdown and harbored at least one FOXA1 binding site in their promoter regions, thereby representing potential common targets co-regulated by MGAT4EP and FOXA1. Finally, through an analysis of TCGA data, we found that four of the 19 candidates, including FOXM1, MALRD1, RMI2, and TGFB3, showed a significant upregulation in the luminal A breast cancer compared with normal breast tissues (log$_2$Fold-Change≥1 and FDR < 0.05). Given that FOXM1 is an established oncogenic transcription factor [47] which is upregulated in a variety of human cancers, we focused on characterizing the mechanism, whereby MGAT4EP/FOXA1 axis regulates its expression.

Both FOXA1 and FOXM1 showed a significant upregulation in luminal A breast cancer subtype compared with normal breast tissues (Additional File 2: Fig. S4C). To validate the regulation of FOXM1 expression by MGAT4EP, we assessed the effect of sgRNA-mediated depletion of MGAT4EP on FOXM1 expression and found that sgRNA-mediated knockdown of MGAT4EP significantly reduced FOXM1 expression at both RNA and protein level in MCF7 cells (Additional File 2: Fig. S4D and E). Consistent with the results from CRISPRi-based silencing, siRNA-mediated depletion of MGAT4EP reduced FOXM1 expression at both RNA and protein level in MCF7 and T47D cells (Fig. 5B, C). In addition, effective siRNA-mediated depletion of FOXA1 markedly reduced FOXM1 expression at RNA and protein level in MCF7 and T47D cells (Fig. 5D, E). These results confirmed that FOXM1 as a common downstream target of FOXA1 and MGAT4EP. We also found that siRNA-mediated depletion of MGAT4EP did not affect the protein level of FOXA1 (Additional File 2: Fig. S4F), indicating that MGAT4EP did not regulate FOXM1 expression by impacting the FOXA1 protein level. Given the previous findings that long noncoding RNAs (lncRNAs) can promote the recruitment of epigenetic modifiers to specific genomic locations [48], we hypothesized that unitary pseudogene MGAT4EP may regulate FOXM1 expression by enhancing FOXA1 binding to its promoter region. To test this hypothesis, we evaluated the effect of siRNA-mediated depletion of MGAT4EP on FOXA1 binding to the promoter region of FOXM1 by ChIP-qPCR. Based on the FOXA1 binding site in the FOXM1 promoter that was identified using publicly available ChIP-seq data [46] (Fig. 5F), we found that FOXA1 bound to FOXM1 promoter and its binding was indeed impaired upon siRNA-mediated depletion of MGAT4EP (Fig. 5G). To determine the role of FOXM1 in luminal A breast cancer cells, we examined its loss-of-function phenotype in MCF7 and T47D cells with short hairpin RNA (shRNA)-mediated gene silencing. Consistent with the oncogenic role of FOXM1 in other cancers, the effective depletion of FOXM1 (Fig. 5H) by either gene-specific shRNA inhibited the growth of MCF7 and T47D cells (Fig. 5I) and impaired their clonogenic capacity (Additional File 2: Fig. S4G).

## Many unitary pseudogenes show clinically relevant expression patterns in human cancer

Unitary pseudogenes are a special class of pseudogenes that derive from acquisition of disrupting mutations in functional protein-coding genes, without duplication or

retrotransposition events. They do not have functional protein-coding gene counterparts in the same genome. With the discovery of MGAT4EP, a novel functional cancer-testis unitary pseudogene, we further investigated among 170 annotated unitary pseudogenes (GENCODE V22), whether there are other unitary pseudogenes, the expression of which is elevated in tumor compared with the corresponding normal tissues and/or is associated with clinical outcomes including patient overall survival (OS) and/or relapse-free survival (RFS) across different types of cancers, via integrative analyses of TCGA data. Interestingly, the majority of unitary pseudogenes showed a significant differential expression ($|\log_2\text{Fold-Change}| \geq \log_2(1.5)$, $p < 0.05$ and FDR $< 0.25$) between tumors and the corresponding normal tissues (Fig. 6A and Additional File 6: Table S5), with 132 unitary pseudogenes showing a significant up-/downregulation in at least three cancer types. In addition, the expressions of 111 and 34 unitary pseudogenes were associated with OS and/or RFS of patients (Fig. 6B, C and Additional File 6: Table S5) respectively in at least two cancer types, based on multivariate Cox proportional hazards regression analysis ($p < 0.05$ and FDR $< 0.25$).

Some examples of unitary pseudogenes, whose expression was significantly associated with OS and/or RFS in the same cancer type or different cancer types, are of particular interest. The first example is CMAHP (cytidine monophospho-N-acetylneuraminic acid hydroxylase, pseudogene), a unitary pseudogene that encodes a dysfunctional version of the cytidine monophospho-*N*-acetylneuraminic acid hydroxylase (Cmah) from other mammals. The enzyme encoded by Cmah in non-human mammals hydroxylates N-acetylneuraminic acid (Neu5Ac), producing N-glycolylneuraminic acid (Neu5Gc) [49]. Neu5Ac and Neu5Gc are two most common forms of sialic acid in many non-human mammals. In contrast, Neu5Gc is not detectable in normal human tissues and is immunogenic in human [49]. Higher CMAHP expression was associated with better patient OS in lung adenocarcinoma (LUAD, log-rank test, $p = 0.00142$)) and cutaneous melanoma (SKCM, log-rank test, $p = 7.47 \times 10^{-6}$), respectively (Fig. 6D). It was also significantly downregulated in these two cancer types compared with the corresponding normal tissues (data not shown), suggesting its tumor-suppressive role. The second example is CPHL1P (ceruloplasmin And Hephaestin Like 1, pseudogene). Different from CMAHP, higher expression of unitary pseudogene CPHL1P was associated with worse patient OS (log-rank test, $p = 4.98 \times 10^{-6}$) in clear cell renal cell carcinoma (KIRC) and worse patient RFS (log-rank test, $p = 0.00245$) in prostate cancer (PRAD), respectively (Fig. 6E). It was also significantly upregulated in KIRC and PRAD compared with the corresponding normal tissues (data not shown), suggesting its tumor-promoting role. The third example is MYH16 (myosin heavy chain 16 pseudogene). Like MGAT4EP, MYH16 is a cancer-testis unitary pseudogene, which was upregulated in solid tumors such as pancreatic adenocarcinoma (PAAD) and showed a much higher expression in testis than other normal tissues (data not shown). It encodes a deficient sarcomeric myosin heavy chain that is otherwise expressed and functional in non-human primate masticatory muscles [50]. The pseudogenization of the sarcomeric myosin heavy chain in human lineage was associated with the marked size reductions in individual muscle fibers and entire masticatory muscles in human [50]. Interestingly, a higher MYH16 expression was associated with both worse patient OS (log-rank test, $p = 0.000454$) and RFS (log-rank test, $p = 0.00363$) in PAAD (Fig. 6F).

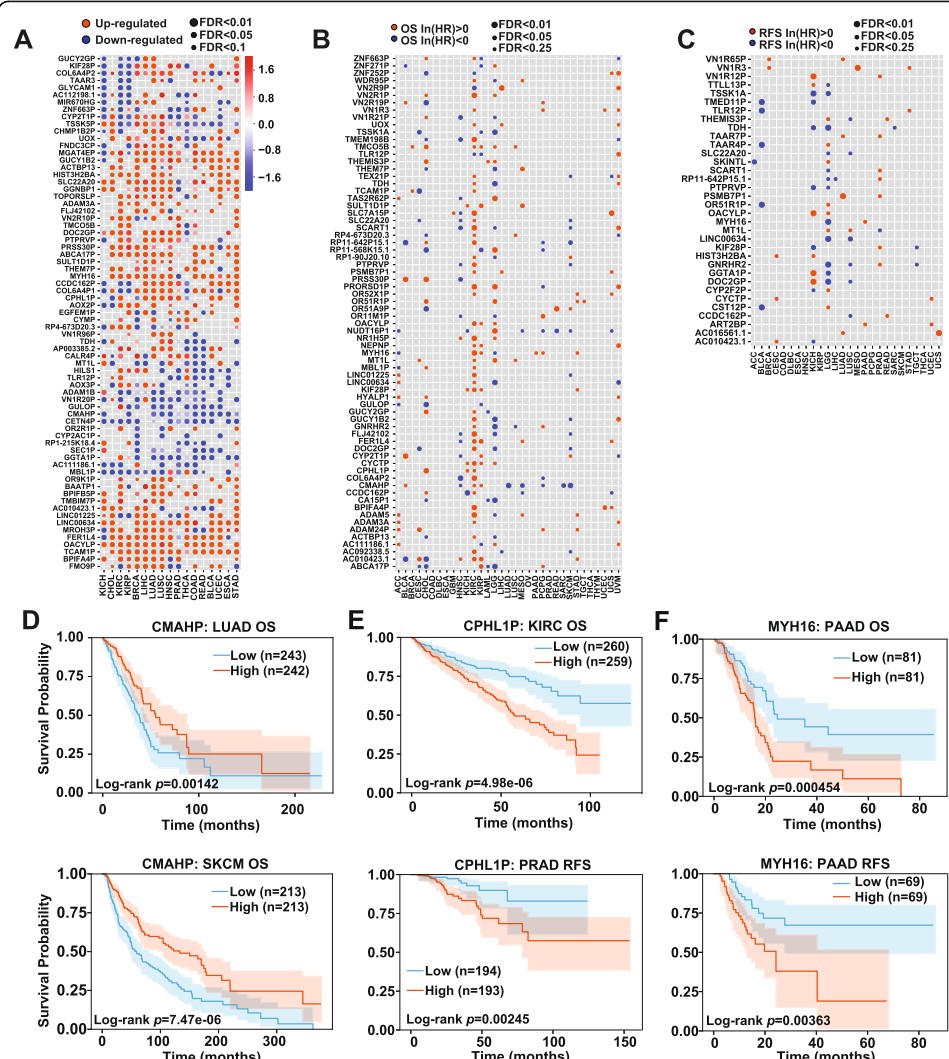

**Fig. 6** Integrative analyses of TCGA data reveal clinically relevant unitary pseudogenes in human cancer. **A** The unitary pseudogenes with significant up-/downregulation in tumors vs. the corresponding normal tissues in at least four cancer types, based on TCGA data are shown. The circle size is proportional to the significance level and the log$_2$(Fold-Change) between tumors and normal tissues is shown by color scale. The unitary pseudogenes, whose expression was significantly **B** associated with patient OS in at least three cancer types or **C** associated with patient RFS in at least two cancer types are shown. The circle size is proportional to the significance level. The unitary pseudogenes, whose expression showed positive and negative natural logarithm of hazard ratio (HR) in a given cancer type, is colored in red and blue, respectively. **D** Higher CMAHP expression was associated with better patient OS in LUAD and SKCM, respectively. **E** Higher expression of unitary pseudogene CPHL1P was associated with worse patient OS in KIRC and worse patient RFS in PRAD, respectively. **F** Higher MYH16 expression was associated with both worse patient OS and RFS in PAAD. The *p* values shown in the figures were calculated based on log-rank test. The Kaplan-Meier survival curves are plotted as solid lines accompanied by 95% confidence interval

## Discussion

With the advancement of computational methods for cataloging pseudogenes in the past decades, it is evident that pseudogenes are present in the genomes of both pro-karyotic and eukaryotic species and are often lineage-specific [4]. Mammalian genomes (e.g., mouse, macaque, and human) encode about ten times more pseudogenes than those from non-mammalian metazoans (e.g., worm, fly, and zebrafish), suggesting that pseudogenes play an important role in mammals [4]. Consistent with this observation,

recent studies revealed that a subset of pseudogenes in mammals exert important regulatory function [6–10]. However, function of the vast majority of pseudogenes are unknown, due to a significant delay in the functional investigation of pseudogenes compared with the fast pace of cataloging pseudogenes. Large-scale pooled screen with RNAi-based technique has facilitated the discovery of protein-coding gene function, but it is difficult to apply RNAi-based loss-of-function approaches to study pseudogenes because of a high sequence homology in transcript sequence between pseudogenes and their parent genes. To enable a large-scale interrogation of human pseudogene function, we leveraged the CRISPRi functional genomics platform and developed an integrated computational pipeline that combines FANTOM5 CAGE data, GENCODE transcriptome annotation, and a supervised algorithm, SCC [23], to design effective sgRNAs to specifically repress the expression of individual pseudogenes at a transcription level.

From the pseudogene-focused pooled CRISPRi screen in MCF7 cells, we identified 69 pseudogenes that were critical for cell fitness. In addition, parent genes generally showed a stronger loss-of-function phenotype than their corresponding pseudogenes, suggesting that in overall, parent genes may be functionally more important than their pseudogene counterpart. Interestingly, upon silencing, a subset of pseudogenes showed a stronger growth inhibitory effect than their parent genes. This finding suggested that some pseudogenes might exert function through pathways independent of their parent genes, which awaits further studies. Genome-wide CRISPR-Cas9 knockout screens across a large panel of cancer cell line models reveal that many protein-coding genes affects cell fitness in a cell-type/cell line-specific manner [29]. Therefore, it remains to be determined the functional pseudogenes identified from our study in luminal A breast cancer cells are commonly essential or specifically critical for the fitness of certain cell types/cell line models. Extending our proof-of-principle screens to a large panel of cell line models to identify the common or context-specific functional pseudogenes will be an important next step. Our in silico analysis of potential CRISPRi off-target effect suggests that the putative off-target effect might have little impact on the results of our CRISPRi screens. One caveat of our analysis is that it was based on the established knowledge about the off-target effect of CRISPR-Cas9 system that involves genome cutting, whereas CRISPRi is based on catalytically inactive Cas9 and does not involve genome cutting. Given our limited knowledge about CRISPRi-mediated off-target effect, the predictive power of such in silico analysis remains unclear. Therefore, it is critical to experimentally validate individual pseudogene hits identified from CRISPRi screens, using function rescue experiments and alternative loss-of-function approaches, as we did in the current study. It will also be important to develop high-throughput assays similar to the ones employed for CRISPR-Cas9 system to better understand, model, and predict CRISPRi-mediated off-target effect on a genome scale.

Among the top pseudogene hits, we identified a unitary pseudogene that was dominantly expressed in testis among normal tissues and was hijacked by cancer cells, as part of an oncogenic transcriptional regulatory circuitry that involved transcription factor FOXA1 and FOXM1, to promote cancer cell fitness. This represents the first cancer-testis unitary pseudogene that plays a tumor-promoting role, via a novel nuclear function different from traditional function mechanisms of pseudogene in cytoplasm.

Unlike other classes of pseudogenes, unitary pseudogenes derive from the lineage-specific acquisition of disrupting mutations in functional protein-coding genes, without duplication or retrotransposition events. As a result, unitary pseudogenes do not have functional protein-coding gene counterparts in the same genome, but only have functional protein-coding orthologs in the genome of other organisms. Because an mRNA may have protein- (coding-dependent) and RNA-mediated (coding-independent) function and unitary pseudogenes lose coding-dependent function of ancestor mRNAs, unitary pseudogenes provide unique opportunities to study how pseudogene function evolves from the RNA-mediated function of its ancestral orthologous mRNAs in other organisms. A previous study [8] revealed that *Pbcas4*, a mouse unitary pseudogene that lost its protein-coding capability during rodent evolution, can function as a ceRNA that regulates the expression of other protein-coding genes. It does so by preserving the key miRNA target sites from the 3′ UTR of its protein-coding ortholog BCAS4 in human and regulates the expression a set of mouse protein-coding genes orthologous to the ceRNA targets of BCAS4 in human. This example demonstrates a conserved unitary pseudogene function that is inherited from the RNA-mediated function of its ancestral ortholog mRNA. In contrast, our study revealed a distinct evolutionary scenario, where the MGAT4EP showed a novel function in transcriptional regulation and dominant nuclear localization that was hijacked by cancer cells to increase cell fitness. Our finding suggests that during evolution, a unitary pseudogene may acquire new RNA-based function which is distinct from that of its ancestral ortholog mRNA. The evolutionary mechanism of how MGAT4EP acquires novel nuclear function remains unclear and thus warrants further investigation. It will be also important to determine in the future studies, the prevalence of unitary pseudogenes exhibiting distinct versus conserved RNA-based function from their ancestral mRNAs. Compared with other classes of pseudogenes, the functional studies of unitary pseudogenes are rather limited. To further explore the potential role of unitary pseudogenes in complex diseases such as cancer, we performed an integrative genomic analysis of TCGA data. Among 170 annotated unitary pseudogenes (GENCODE V22), we identified 132 of them that showed significantly differential expression between cancers and the corresponding normal tissues in at least three cancer types. Moreover, there were 111 and 34 unitary pseudogens, the expression of which was found to be associated with OS or RFS of patient, in at least two cancer types. Thus, like MGAT4EP, many other clinically relevant unitary pseudogenes may play an important role in human cancer, and further efforts are needed to characterize their functions.

## Conclusions

Our proof-of-principle study represents the first large-scale systematic effort for characterizing pseudogene function. Given that the pseudogenes exist in the genome of a wide variety of organisms, and the general utility of CPRISPRi technique in both prokaryotes and eukaryotes [18, 19], large-scale studies like the current one promise to open new avenues for revealing the role of pseudogene in different biological contexts across evolutionarily distant species. The combination of functional genomics studies and detailed mechanistic ones may ultimately provide insight into how pseudogenes have become an integrated functional component of cellular circuitry over the course of evolution. Our findings suggest the importance of newly evolved nuclear function of

unitary pseudogenes and underscore their underappreciated roles in human diseases. Finally, the CRISPRi sgRNA library designed in the current study for inhibiting human pseudogene expression will serve as a useful resource for facilitating the characterization of human pseudogene function under different physiological and pathological conditions.

## Methods

### TCGA and CCLE data analysis

The GENCODE V22 annotation was retrieved from https://www.gencodegenes.org/. The RNA-seq data and clinical information for TCGA breast cancer cohort [36] were downloaded from GDC Data Portal. The genes with deregulated expression between all breast tumors/PAM50 breast cancer subtypes [24] and normal tissues were identified using edgeR [51] (3.24.3) with the filters of $|\log_2\text{Fold-Change}| \geq \log_2(1.5)$ and FDR < 0.05, based on the uniquely mapped RNA-seq reads. The unitary pseudogenes showing differential expression between tumor and normal tissues in different cancer types were also identified using edgeR [51] (3.24.3) with the filters of $|\log_2\text{Fold-Change}| \geq \log_2(1.5)$, $p < 0.05$ and FDR < 0.25, based on summarized RNA-seq read count data from TCGA. Only the unitary pseudogenes that are expressed in at least one cancer type with the corresponding normal tissue in TCGA were included for differential expression analysis. The unitary pseudogenes whose expression showing significant correlation with overall survival and/or relapse-free survival was identified using the multivariate Cox proportional-hazards regression analysis that included both clinical variables (i.e., age, gender, tumor stage, and/or grade) and pseudogene expression, with the filters of $p < 0.05$ and FDR < 0.25. For individual unitary pseudogenes that passed the statistical significance threshold of multivariate Cox model, the log-rank test and the Kaplan-Meier method were used for analyzing their survival data. The CCLE breast cancer cell line RNA-seq data was downloaded from GDC Data Portal. The raw sequencing reads were mapped to the hg38 genome and GENCODE V22 transcriptome using HiSAT2 [52] with parameters "--no-discordant --no-mixed." To quantify pseudogene expression in cell lines, the uniquely mapped RNA-seq read counts were generated using featureCounts [53]. We defined a pseudogene version of FPKM as (gene-level read count $\times 10^{-9}$)/(effective length $\times$ total read count), in which the effective length of a gene was defined as its genomic length that is uniquely mappable with a defined read length.

### CRISPRi sgRNA library design and construction

FANTOM5 cap analysis of gene expression (CAGE) data was integrated with GEN-CODE V22 transcriptome annotation to define the transcription starting sites (TSSs) of protein-coding genes, lncRNAs, and pseudogenes, as described previously [22]. The genomic sequences within the 500-base pair (bp) window centered on each TSS were used for sgRNA design. The sgRNA design was performed using the Sequence Scan for CRISPR (SSC) method, as described previously [27]. The designed sgRNAs that meet one of the following criteria: (a) being mapped to multiple genomic regions; (b) with any Ns or more than three consecutive T; (c) with extreme level of GC content ($\geq$ 75% or < 10%), were filtered out of the library. If several sgRNAs were within 4 bp from

each other, only the one with the best SSC scores was kept. At gene level, up to 10 top-ranked sgRNAs were selected from the corresponding CAGE-defined TSSs. If multiple CAGE clusters were assigned to a given gene, the sgRNAs were preferably selected from the CAGE clusters with higher transcription initiation evidence score (TIEScore) [22]. The pseudogenes with FPKM ≥ 0.5 in either MCF7 or MDA231 cell line [25] and having at least three designed sgRNAs (5703 sgRNAs targeting 850 pseudogenes) were included in the CRISPRi screen. The parent genes [26] for the selected pseudo-genes were included in the screen if they have at least three designed sgRNAs (3727 sgRNAs targeting 380 parent genes). The 568 sgRNAs targeting 71 core essential genes [27] were included as positive controls, and the 267 sgRNAs targeting AAVS1 and 83 non-targeting sgRNAs were included as negative controls, respectively. In addition to 10,348 sgRNAs designed for the screen in breast cancer cell lines, 1567 sgRNAs used for another unpublished screen were included in the final sgRNA library, which resulted in a total of 11,915 sgRNAs. The flanking linker sequences (5′ linker: CTTTATATATCTTGTGGAAAGGACGAAACACCG; 3′ linker: GTTTTAGAGCTAGAAATAGCAAGTTAAAATAAGGCTAGTCCG) were added to each designed sgRNA sequence for library construction. The oligonucleotides containing both sgRNAs and flanking linker sequences were synthesized as a pooled library using the CustmoArray 12K chips (CustmoArray, Inc). The array-synthesized sgRNA library was amplified for 8 cycles (primer sequences in Additional File 4: Table S3) with Q5 High-Fidelity DNA Polymerase (New England Biolabs #M0491S). The PCR product was purified from 2% agarose gel with QIAquick Gel Extraction Kit (QIAGEN # 28704). Gibson assembly (Gibson Assembly® Master Mix, New England Biolabs # E2611L) was used to assemble the amplified sgRNA library into a BsmBI (Thermo Fisher # ER0452)-digested lentiGuide-Puro vector (Addgene #52963). A total of 2 μl of 10–50 ng/μl product from Gibson assembly reaction was added to one tube of 25 μl electrocompetent cells (Lucigen) on ice for 5 min (∼ 3–4 reactions for one library). Electroporation was then conducted using Micropulser Electroporator (Bio-Rad) by one-shot EC1 program. The transformed electrocompetent cells were recovered in recovery media and was rotated at 250 rpm for 1 h at 37 °C. One milliliter of transformation was plated on each of pre-made 24.5 cm$^2$ bioassay plates (ampicillin) using a spreader. All plates were grown inverted for 14 h at 32 °C. Finally, the colonies were scraped off and the plasmids were extracted with NucleoBond Xtra Midi EF kit (Takara # 740422.50) for downstream virus production.

### Cell culture

The MCF7 cell line with a stable expression of (Sp) dCas9-KRAB fusion protein (MCF7-dCas9) was a gift from Dr. Howard Y. Chang's laboratory at Stanford. The MCF7, MCF7-dCas9, and 293FT cell lines were cultured in Dulbecco's modified Eagle's medium (DMEM, Hyclone #SH30022.01), supplemented with 10% fetal bovine serum (FBS, Gibco #10437-028), and 1% penicillin/streptomycin (Corning #30-002-CI). The authenticated MCF7 and 293FT cell lines were obtained from Characterized Cell Line Core facility at MD Anderson Cancer Center (MDACC). The T47D cell line was obtained from the ATCC and cultured in

RPMI-1640 (Hyclone #SH30027.1) supplemented with 10% FBS (Gibco #10437-028) and 1% penicillin/streptomycin (Corning # 30-002-CI). All cell lines were cultured in an incubator (Thermo, HEARCELL VIOS 160i) with 5% $CO_2$ at 37 °C.

## CRISPRi screen and data analysis

Lentiviruses containing sgRNA library were generated by co-transfection of pCMV-VSV-G, psPAX2, and sgRNA library plasmid into 293FT cells. The supernatant containing lentiviruses was collected 48 h post-transfection. MCF7-dCas9 cells were plated into ten 10-cm dishes and infected with lentiviruses containing the sgRNA library at an MOI of 0.2~0.3. After cells were selected with puromycin (2 μg/ml) for 4 days, $4.8 \times 10^7$ cells were split into three replicates. For each replicate, $1 \times 10^7$ were harvested to extract genomic DNA (D0) using QIAamp DNA Mini Kit (QIAGEN), and $6 \times 10^6$ cells (~ 500× coverage for each sgRNA per replicate) were passed every 3 days and cultured for 21 days. At day 21 (D21), $1 \times 10^7$ cells were harvested for each replicate to extract genomic DNA. Two rounds of PCR were employed to prepare the next-generation-sequencing (NGS)-ready sgRNA libraries with the KAPA HiFi HotStart ReadyMix (Roche # KK2602). The first-round PCR was conducted for 16 cycles, using 40 μg of input genomic DNA from each replicate at D0 or D21 as a template. The PCR product of different samples was pooled and 20 μl of the mixed product was used as a template for the second-round PCR, which was conducted for 12 cycles to incorporate Illumina barcode sequences (Forward: AATGATACGGCGACCACCGAGATCTACAC<Illumina index 8-nt barcode > ACACTCTTTCCCTACACGACGCTCTTCCGATCTTCTTGTGGAAAGGACGAAACACCG; Reverse: CAAGCAGAAGACGGCATACGAGAT<Illumina index 8-nt barcode > GTGACTGGAGTTCAGACGTGTGCTCTTCCGATCTCTACTATTCTTTCCCCTGCACTGTACC). The final PCR product was purified from 2% agarose gel with QIAquick Gel Extraction Kit. Concentration of different libraries was measured using the Qubit dsDNA HS (High Sensitivity) Assay Kit (Thermo # Q32851) on a Qubit Fluorometer (Thermo Fisher). The libraries were pooled with equal proportion for NGS (single-end 75 bp) on an Illumina NextSeq 500 system. All primer sequences are listed in Additional File 4: Table S3. The raw sequencing reads were mapped to sgRNA sequences in the library, using Bowtie [54] (1.2.2) with parameters "--best --strata -a --norc -m 1 -5 20 -3 30." Samtools [55] (1.2.0) was used to calculate the read count of individual sgRNAs. MAGeCK [30] (0.5.7) was used to identify the negatively or positively selected sgRNAs and genes, from the sgRNA read count table, with the following parameters: --norm-method control --gene-lfc-method secondbest --control-sgrna negctrl.lst --normcounts-to-file --additional-rra-parameters "--permutation 10000". The filters of $p < 0.05$, FDR < 0.25, and $\log_2$Fold-Change $\leq -\log_2(1.5)$ were used to define the negatively selected hits from the screen. To control for the false positives caused by bidirectional promoters, we excluded the negatively selected genes from screen hits, if their TSSs were within 1 kb from the TSSs of another gene based on GENCODE V22 annotation. The TSS distance between two genes was calculated as the minimum of the distances between the CAGE-based assigned TSSs, and the distances between the annotated TSSs of two genes in GENCODE V22.

## CRISPRi off-target effect analysis

Cas-OFFinder [31] was used to predict the putative genomic off-target sites for individual sgRNAs based on their sequences, with the parameters of PAM Type = SpCas9

from Streptococcus pyogenes: 5′-NGG-3′, mismatch number = 1, DNA bulge size = 0, RNA bulge size = 0. A sgRNA targeting pseudogene/parent gene is considered to have a putative off-target effect on the corresponding parent gene/pseudogene if at least one of its predicted off-target sites is within [− 2 kb, + 1 kb] from the TSS of its corresponding parent gene/pseudogene.

### Real-time quantitative reverse transcription PCR (qRT-PCR)

Total RNA was extracted from MCF7 and T47D cells using the RNeasy Mini kit (QIAGEN #74104), according to the manufacturer's manual. RNA concentration was measured with a NanoDrop spectrophotometer, and 1 μg of total RNA was used for the synthesis of cDNA using the iScript™ Reverse Transcription Supermix (Bio-Rad #1708841). QRT-PCR was performed using SsoAdvanced Universal SYBR Green Supermix (Bio-Rad #1725274) in the CFX96 Touch Real-Time PCR Detection System (Bio-Rad) according to the manufacturer's manual. The sequence of primers used in this study is listed in the Additional File 4: Table S3. Glyceraldehyde 3-phosphate dehydrogenase (GAPDH) was used as an internal control, and the fold change of pseudogene or gene expression was calculated using the ΔΔCT method.

### CRISPRi, RNAi-mediated gene silencing, and pseudogene cDNA overexpression

To validate pseudogene hits identified from the screen using CRISPRi-mediated gene silencing, the top 2 sgRNAs showing the strongest growth inhibitory effect in the CRISPRi screen were selected and cloned into lentiGuide-Puro vector. To produce lentiviruses, HEK293T cells were co-transfected with pCMV-VSV-G, psPAX2, and sgRNA-expressing lentiGuide-Puro plasmid using jetPRIME (Polyplus transfection #114-15). A non-targeting sgRNA or genome-targeting sgRNA was used as a negative control. Lentiviruses were collected 48 h after transfection and were then used to infect cell lines with stable expression of (Sp) dCas9-KRAB fusion protein in the presence of polybrene (Sigma #TR-1003) prior to puromycin selection for 4 days. Total RNA was extracted using RNeasy Mini Kit (QIAGEN) from cells 10 days after lentiviral infection, and qRT-qPCR was performed to determine the knockdown efficiency of individual sgRNAs. For siRNA-mediated knockdown of protein-coding genes, one non-targeting siRNA and two pre-designed on-targeting siRNAs (Sigma-Aldrich) were used. To achieve effective siRNA-mediated knockdown of MGAT4EP transcript, which is predominantly localized in the nucleus, chemically modified gene-specific silencer select siRNAs and non-targeting siRNAs (Thermo Fisher) were used. A total of $1.5 \times 10^5$ cells were plated in each well of 6-well plates. In each well, 100 pmol siRNAs were transfected into cells using Lipofectamine RNAiMAX Transfection Reagent (Thermo Fisher #13778150), and total RNA was extracted 48 h after transfection for qRT-PCR analysis of knockdown efficiency. For pseudogene expression, full-length cDNA sequences of MGAT4EP (NR_038135.2), DDX12P (NR_033399.1), PRELID1P1 (NC_000006.12), and TUBBP5 (NR_027156.1) were synthesized (Twist Bioscience) and inserted into pTwist CMV Puro vector (Twist Bioscience) between NotI and BamHI restriction enzymes sites. The pseudogene expression plasmids were transfected into cells using the Lipofectamine 3000 reagent (Invitrogen #L3000015). For shRNA-mediated knockdown, the shRNA sequences were cloned into PLKO.1 TRC vector. To produce lentiviruses,

HEK293T cells were co-transfected with pCMV-VSV-G, psPAX2, and shRNA-expressing PLKO.1 TRC plasmid using jetPRIME. Lentiviruses were collected 48 h after transfection and were then used for infecting MCF7 or T47D cell lines in the presence of polybrene prior to puromycin selection for 2 days. Total RNA and protein were collected 4 days after infection. QRT-PCR and western blot were used to determine the efficiency of shRNA-mediated knockdown at RNA and protein level, respectively. All sgRNA, siRNA, and shRNA sequences are listed in the Additional file 4: Table S3.

### Cell proliferation and clonogenic assays

To assess the growth inhibitory effect of gene-specific siRNA-mediated silencing, 96 h after siRNA transfection, cells were trypsinized, resuspended, and seeded at 1000 cells per well in a 96-well plate, where each treatment condition and time point was in triplicate. From the following day (day 0) to 4 days afterwards, cell proliferation was assessed using Cell Counting Kit-8 (CCK-8) assay (Dojindo Molecular Technologies #CK04-13). Briefly, 10 μl CCK-8 solution was added into each well. Next, the OD450 absorbance was measured after 2 h incubation at 37 °C. The CCK-8-based proliferation assay was performed similarly for the cells transduced with shRNA/sgRNA, except that cells transduced with shRNA/sgRNA were seeded after 4 or 8 days of puromycin selection. Clonogenic assays were performed as follows. ShRNA/sgRNA-transduced cells were seeded at 1000 cells per well in 6-well plates, with each treatment condition in triplicate. Medium was changed every 4 days. After 2 weeks, cells were fixed with 100% methanol and stained with 0.5% crystal violet in PBS. Plates were then washed with distilled water and photographed with ChemiDoc Touch Imaging Systems (Bio-Rad).

### Nuclear and cytoplasmic fractionation

Nuclear and cytoplasmic RNAs of MCF7 and T47D cells were isolated using the PARIS™ kit (Thermo Fisher # AM1921) according to the manufacturer's manual. Briefly, $5 \times 10^6$ cells were collected and washed with cold PBS and were then lysed with 500 μl ice-cold cell fractionation buffer on ice for 10 min. After centrifugation for 5 min at 4 °C and 500×g. the supernatant containing cytoplasmic fraction and the nuclei pellet were collected, respectively. The collected nuclei pellet was washed with ice-cold cell fractionation buffer and repelleted by centrifugation for 1 min at 4 °C and 500×g, followed by lysis with cell disruption buffer. The nuclear lysate or the cytoplasmic fraction was mixed with an equal volume of 2× lysis/binding solution and 100% ethanol. The mixture was then transferred to a filter cartridge for RNA purification. MALAT1 RNA and GAPDH mRNA were detected by qRT-PCR in isolated nuclear/cytoplasmic RNAs, as a control for nuclear and cytoplasmic RNA, respectively. In addition, β-tubulin and histone H3 protein were detected by western blotting in isolated nuclear/cytoplasmic fractions, as a control for nuclear and cytoplasmic protein, respectively.

### 5′ and 3′ RACE

The 5′ and 3′ RACE experiments were conducted using the SMARTer® RACE 5′/3′ Kit (Clontech #634859). Briefly, the total RNA of MCF7 cells was extracted using the RNeasy Mini kit (QIAGEN #74104) according to the manufacturer's instruction. First-strand cDNA was synthesized using 5′-CDS and 3′-CDS primer A and SMARTer II A

oligonucleotide as described in the user's manual. The touchdown nested PCR was used to amplify cDNA ends. The PCR product was purified from 2% agarose gel with NuceloSpin Gel and PCR Clean-Up Kit (supplied with the SMARTer® RACE 5′/3′ Kit) and was then cloned into pRACE vector using In-Fusion HD Master Mix (both vector and mix were provided as SMARTer RACE 5′/3′ Kit Components). Finally, the clones containing the gene-specific inserts were sequenced.

### Ribo-seq and mass spectrometry data analysis for searching potential MGAT4EP-encoded proteins/micropeptides

The ribo-seq data was analyzed as described previously [40]. Briefly, ribosome-protected RNA fragment (RPF) reads were trimmed and the low-quality reads were filtered by Sickle (http://github.com/ucdavis-bioinformatics/sickle). The RPF reads after filtering were mapped to human rRNA sequences using bowtie and allowing for two mismatches. The reads that were not mapped to human rRNA sequences were then mapped to human genome (GRCh38) with GENCODE V22 annotation using STAR (2.6.1b) [56]. The alignment was performed with the following parameters: "−outSA-Mattributes All−outFilterMismatchNmax 2−alignEndsType EndToEnd−outFilterIn-tronMotifs RemoveNoncanonicalUnannotated−alignIntronMax 20000−outMultim apperOrder Random−outSAMmultNmax 1." For MS data analysis, the customized protein sequence database was constructed by merging the non-redundant protein sequences from Uniprot (release 2019_06) (20431 reviewed human proteins), Ensembl (GRCh38.79) (100778 human proteins), and NCBI RefSeq (GRCh38.v20200819) (114963 human proteins), together with the protein sequences corresponding to the putative ORFs (26 ORFs with ATG start codons and 48 ORFs with non-ATG start codons) identified by an ORF prediction module (ribotish.zbio.orf.allorf) that solely relies on the sequence information and is implemented in Ribo-TISH package [40], based on the MGAT4EP sequence. The reverse sequences of the proteins in the database were used for the target-decoy-based MS/MS spectrum search. The raw MS/MS data generated in MCF7 and T47D cells [41] were first converted into mzML files using MScon-vert (ProteoWizard, version 3.0.20282) [57] and were then searched against our customized protein databases, using MS-GF+ (v2020.08.05) [58]. The following parameters were used for database searching: fixed modifications, Carbamidomethyl (C); variable modifications, Oxidation (M); Precursor ion mass tolerance, 20 ppm; Range of allowed isotope peak errors, "0,0"; Enzyme specificity, trypsin; maximum missed cleavages. The target-decoy approach implemented in PGA [59] was used to estimate the FDR with the module "separate FDR estimation." All the results were filtered with 1% FDR at a peptide level.

### In vitro translation assay

The in vitro translation assays were performed by using TnT® T7 Quick Coupled Transcription/Translation System (Promga, Cat.# L1170), according to the manufacturer's instructions. The DNA template of MGAT4EP for in vitro translation was prepared by PCR amplification via adding T7 RNA polymerase promoter to the 5′ end of MGA-T4EP full-length cDNA. The proteins generated from in vitro translation reactions

were further detected using Transcend™ Colorimetric Translation Detection System (Promega, Cat.#L5072) according to the manufacturer's instructions.

### RNA pull-down and RNA immunoprecipitation (RIP)

To isolate proteins that interact with MGAT4EP transcript (Refseq NR_038135.2), we used Pierce™ Magnetic RNA-Protein Pull-Down Kit (Thermo Fisher #20164) and adopted a published protocol, as described previously [44]. Briefly, the full-length MGAT4EP cDNA sequence was PCR-amplified with an addition of T7 promoter sequence to its 5′ end (see primer sequences in Additional File 4: Table S3). T7 Ribo-MAX™ Express Large-Scale RNA Production System (Promega #P1320) was used to produce full-length, antisense, and deletion mutants of MGAT4EP RNAs by in vitro transcription with the PCR-amplified T7-promoter-cDNA template, according to the manufacturer's instruction. Transcribed RNAs were purified using RNeasy Mini Kit (QIAGEN) and desthiobiotin-labeled using Pierce RNA 3′ End Desthiobiotinylation Kit (Thermo Fisher). A total of 50 pmol desthiobiotin-labeled RNA was incubated with 50 μl streptavidin magnetic beads for 30 min at room temperature with agitation. RLN buffer and protein lysis buffer were used to prepare nuclear fraction extract from MCF7 cell, as described previously [44]. The streptavidin magnetic beads were then washed twice with an equal volume of 20 mM Tris buffer and incubated with prepared nuclear fraction extract in protein-RNA binding buffer at 4 °C with agitation or rotation for 1 h. After washing 4 times with wash buffer, the RNA-binding protein complexes were eluted with elution buffer and analyzed with mass spectrometry or western blot.

The RIP assay was conducted following the manufacturer's manual using the EZ-Magna RIP™ RNA-Binding Protein Immunoprecipitation Kit (Millipore). Briefly, cells in a 15-cm plate was washed with ice-cold PBS, scraped off from each plate, and collected by centrifugation at 1500 rpm for 5 min at 4 °C. Collected cell pellet was resuspended in an equal pellet volume of complete RIP Lysis Buffer, incubated on ice for 5 min, and stored at − 80 °C. Next, the magnetic beads were washed with RIP wash buffer and incubated with antibodies for 30 min at room temperature with rotation. After incubation, the antibodies-beads complex was washed twice with RIP wash buffer. Once thawed, the RIP lysate was centrifuged at 14,000 rpm for 10 min at 4 °C. One hundred microliters of the supernatant was mixed with antibody-beads complex in RIP immunoprecipitation buffer, and the mixture was incubated at 4 °C for 4 h with rotating. The beads were washed 6 times with RIP wash buffer and then incubated with proteinase K at 55 °C for 30 min with shaking to digest the protein. Finally, RNA was extracted with phenol-chloroform for qRT-PCR analysis.

### RNA-seq experiments in cell line and data analysis

Total RNA was prepared from MCF7 cells using RNeasy Mini Kit (QIAGEN) and was treated with DNase I (QIAGEN #79254). Two micrograms of RNA was used for RNA-seq library construction with TruSeq Stranded mRNA Library Prep kit (Illumina # 20020594). Sequencing of the library (75 bp single-end read) was conducted on an Illumina NextSeq 500 System, at the Advanced Technology Genomics Core of MDACC. The RNA-seq reads were trimmed for adaptor sequence and masked for low-complexity and low-quality sequence. They were then mapped to the hg38 genome and

GENCODE V22 transcriptome, using STAR (2.6.1b) [56] with parameters "--outSA-Munmapped Within --outFilterType BySJout --twopassMode Basic --outSAMtype BAM SortedByCoordinate." The gene-level raw read counts were calculated using htseq-count function of HTSeq (0.11.0) [60] with parameters "--stranded reverse --additional-attr gene_name gene_type." The normalization of raw read counts and differential gene expression between the treatment and control conditions were identified, using DESeq2 (1.22.2) [61] ($|\log_2$Fold-Change$| \geq \log_2 1.5$ and FDR $< 0.05$).

### Chromatin immunoprecipitation (ChIP) coupled with quantitative PCR (qPCR)

ChIP assays were conducted using EZ-Magna ChIP™ A/G Chromatin Immunoprecipitation Kit (Millipore # 17-10086), as described in the manufacturer's manual. In brief, MCF7 or T47D cells cultured in 10 cm dish were fixed with 1% formaldehyde for 10 min at room temperature. Reaction was quenched with 125 mM Glycine for 5 min. The cells were scrapped and spun down by centrifugation (Thermo Scientific ST16R Refrigerated Centrifuge). After cell lysis, released nuclei were lysed and the resultant nuclear lysate was subject to sonication with Bioruptor sonication device (Diagenode #B01020001) for DNA fragmentation. The sonication was conducted using the following conditions: 60 cycles of 30 s on and 30 s off at a high level. After sonication, the supernatant was collected and divided into aliquots for immunoprecipitation (IP). Five micrograms FOXA1, pol II or IgG antibody, and 20 μl A/G magnetic beads were incubated with the collected supernatant for overnight at 4 °C (4) to capture crosslinked protein/DNA complex, followed by pelleting magnetic beads with the magnetic separator. The crosslinked protein/DNA complex was then eluted from the magnetic beads and crosslinks of protein/DNA complexes were reversed to free DNA. Finally, the eluted DNA was purified and subject to qPCR analysis. All primer sequences and antibody information are listed in Additional File 4: Table S3.

### ChIP-Seq data analysis

The FOXA1 ChIP-seq data in MCF7 (GSM798439, GSE32222) [46] and T47D (GSM631473, GSE25710) cell lines were downloaded from GEO. Bowtie [54] (1.2.2) was used to map the raw reads to the human hg38 genome with parameters "-S --best −strata -a -m 1". MACS2 [62] (2.1.1) was used to call peaks from mapped reads with parameters "-g hs --call-summits -q 0.05." The identified peaks were considered to be associated with the promoter region (− 1.5 kb to + 500 bp of the TSS) of a given target, if there is at least one bp overlap between the peak and the promoter region.

### Western blot

Total protein extract was prepared from the cultured cell lines using RIPA lysis and extraction buffer (Thermo Fisher #89900) supplemented with protease Inhibitor Cocktail (Sigma #11697498001). The concentration of total protein was quantitated using the Bradford dye-binding method (Bio-Rad # 5000006). Twenty micrograms of protein was loaded and separated by 4–15% Mini-PROTEAN TGX precast polyacrylamide gel (Bio-Rad #4561085), and then transferred to 0.22 μm polyvinylidene fluoride (PVDF) membranes (Millipore # ISEQ00010). PVDF membranes were blocked with 5% non-fat milk and incubated with specific antibodies for detecting different proteins (see detailed

information about antibody information in Additional File 4: Table S3). After the blot is incubated in ECL chromogenic substrate (Millipore # WBKLS0100), protein bands were detected by ChemiDoc Touch Imaging System (Bio-Rad) and the signal was quantified using Image lab software (Bio-Rad).

### Statistical analysis

All the experimental data are presented as the mean ± standard deviations (SD), and the two-tailed Student's *t* test was used to assess the statistical significance between two groups using GraphPad Prism 8.0.

## Supplementary Information

---

**Additional file 1: Supplementary Table 1.** The sequences of the sgRNAs that were designed to target pseudogenes, parent genes as well as positive and negative control sgRNAs.

**Additional file 2: Supplementary Figure 1-7.**

**Additional file 3: Supplementary Table 2.** The CRISPRi screen results for individual pseudogenes and parent genes in MCF7 cells, including the positive and negative selection information about pseudogene/parent genes as well as the significant negatively selected sgRNAs that may have putative off-target effect on pseudogene/parent gene hits.

**Additional file 4: Supplementary Table 3.** The sequences of the sgRNAs, siRNAs, shRNAs and PCR primers and the information about antibodies.

**Additional file 5: Supplementary Table 4.** The list of differentially expressed protein-coding genes between MCF7-dCas9 cells transduced with non-targeting sgRNA control and MGAT4EP-targeting sgRNA.

**Additional file 6: Supplementary Table 5.** The results of differential expression analysis between tumors and the corresponding normal tissues and multivariate Cox proportional hazards regression analysis of association between gene expression and clinical outcomes including patient OS and RFS for all unitary pseudogenes annotated in GENCODE V22, across different cancer types.

**Additional file 7.** Review history.

---

### Acknowledgements

We thank Drs. Seung Woo Cho and Howard Chang for kindly sharing with us the stable cell line of MCF7-dCas9.

### Review history

The review history is available as Additional file 7.

### Peer review information

### Authors' contributions

M.S. and Y.C. designed the study and analyzed the data; M.S., C.Z., J.H., X.L, Y.D., and X.C. conducted the experiments; Y.W., YJ.W., P.Z., W.H., and H.L. performed bioinformatics analysis; C.H. developed and applied the computational pipeline to determine the TSS for individual pseudogenes or parent genes. M.S. and Y.C. wrote the manuscript with the input from other co-authors. H.X. contributed to developing and applying computational pipeline for sgRNA design and co-supervised the study with Y.C. The author(s) read and approved the final manuscript.

### Authors' information

Ming Sun, Yunfei Wang, and Caishang Zheng contributed equally to this work. Correspondence and requests for materials should be addressed to Yiwen Chen (ychen26@mdanderson.org) or Han Xu (HXu4@mdanderson.org).

### Funding

This work was partially supported by the grants from NIH (R01GM130838, R01NS117668 to Y.C. and R35GM137927 to H.X.), Cancer Prevention Research Institute of Texas (RR140071 to Y.C. and RR160097 to H.X.), and Bristol-Myers Squibb-MRA Young Investigator Award in Immunotherapy (#569414 to Y.C.). X.C. has been supported by NIH (R37CA228304, P50CA186784, and R01HL146642), DOD/CDMRP (W81XWH1910524), and American Cancer Society (RSG-18-181-01-TBE). Y.C., H.X., and X.C. are CPRIT Scholar in Cancer Research. X.L. has been supported by DOD/CDMRP (W81XWH-19-1-0035).

### Availability of data and materials

The raw sequencing data generated and/or analyzed in the current study were deposited to GEO (GSE155510) [63]. The code to reproduce all the analyses presented in the paper is available on GitHub (https://github.com/tsznxyz/BrcaPseudoCode) [64] and deposited on Zenodo (https://doi.org/10.5281/zenodo.5148874) [65].

## Declarations

### Ethics approval and consent to participate
Not applicable

### Consent for publication
Not applicable

### Competing interests
The authors declare that they have no competing interests.

### Author details
[1]Department of Bioinformatics and Computational Biology, The University of Texas MD Anderson Cancer Center, Houston, TX 77030, USA. [2]Present affiliation: Department of Oncology Center, The Affiliated Suzhou Hospital of Nanjing Medical University, Suzhou Municipal Hospital, Gusu School, Baita west road #16, Suzhou 215001, China. [3]Present affiliation: Clinical Science Lab, H. Lee Moffitt Cancer Center & Research Institute, Tampa, FL 33612, USA. [4]Present affiliation: Department of Biology and Biochemistry, University of Houston, Houston, TX, USA. [5]Present affiliation: Key Laboratory of RNA Biology, Center for Big Data Research in Health, Institute of Biophysics, Chinese Academy of Sciences, Beijing 100101, China. [6]Department of Epigenetics and Molecular Carcinogenesis, The University of Texas MD Anderson Cancer Center, Houston, TX 77030, USA. [7]Genetics and Epigenetics Program, MD Anderson Cancer Center UTHealth Graduate School of Biomedical Sciences, Houston, TX 77030, USA. [8]Quantitative Sciences Program, MD Anderson Cancer Center UTHealth Graduate School of Biomedical Sciences, Houston, TX 77030, USA. [9]Department of Molecular and Cellular Biology, Baylor College of Medicine, Houston, TX 77030, USA. [10]Lester and Sue Smith Breast Center, Baylor College of Medicine, Houston, TX 77030, USA. [11]Dan L. Duncan Cancer Center, Baylor College of Medicine, Houston, TX 77030, USA. [12]Department of Systems Biology, The University of Texas MD Anderson Cancer Center, Houston, TX 77030, USA. [13]Center for Integrative Medical Sciences, RIKEN, 1-7-22 Suehiro-cho, Tsurumi-ku, Yokohama, Kanagawa 230-0045, Japan.

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

## 
