## [**Additional file 7.** Review history. · Genome Biology]

Review History

First round of review

Reviewer 1

Are you able to assess all statistics in the manuscript, including the appropriateness of statistical tests used? Yes, and I have assessed the statistics in my report.

Comments to author:

The author present results from the first CRISPRi screen performed to date for interrogating human pseudogenes' function.

They devised an effective bioinformatics pipeline to design a library of single guide RNA targeting pseudo genes, combining a CAGE analysis to identify transcription starting sites at a genome scale (including TSS of pseudogenes) with Sequence scan for CRISPR to identify possible target regions for sgRNAs. Secondly, they selected two breast cancer cell lines and focused only on pseudogenes targeted by at least 3 sgRNAs and expressed at the basal level in the two models (using public available RNA-seq data from the cancer cell line aencyclopedia). The final library included also guides targeting pseudo-gene parent genes, plus positive and negative controls, i.e. core-fitness genes and non targeting guides.

Using previously described experimental setting, the authors screened the two selected models with their library and were able to identify a number of pseudo-gene hits, that upon inactivation reduce cellular fitness.

They selected the top hits and put them forward to further experimental validation, functional characterisation and clinical relevance estimation. Among these, they extensively characterise the role of MGAT4EP as a cancer-testis unitary pseudogene promoting cellular fitness in breast cancer.

Briefly, this is a great piece of work describing an unprecedented effort, setting standards for future studies employing CRISPRi screens to characterise pseudo genes and reporting results of potential impact.

Particularly, the bioinformatics pipeline used to design the library is well thought, comprehensively described and the author included appropriate positives and negative controls.

A number of points should be addressed and others clarified, in my opinion before proceeding with the publication of this manuscript in genome biology:

- The authors should mention that the final library is specific to the two screened models (as they excluded pseudogenes not expressed in these two models)
- What is the final median number of sgRNA per pseudogenes? an histogram with n. sgRNA per pseudo genes should be included as a supplementary figure.
- Among the parent genes included in the library is there any MCF7 or MDA-MB-231 specific essential gene? if so these could be used as further positive controls. The authors might use

public available data from genome wide CRISPR-Cas9 recessive screens, available on project score (PMID: 33068406) and the cancer dependency map portal (<https://depmap.org/portal/>), to this aim.

- The authors do not specify if the screens were performed in duplicated batches and how reproducibility across these was assessed.

- The authors should discuss the rationale beyond the selection of the two models employed in their screen and also mention in title, abstract and introduction that their results are specific to breast cancer (they might actually be model specific, i.e. linked to genomic background of MCF7 and MDA-MB-231, this should be mentioned in the discussions)

- the legends of the supplementary figures are sloppy and including repetitions/errors, I would suggest a proper revision. Furthermore what is the difference between supplementary figure 1B and 1C ? Finally, effects on fitness are hardly readable if plotting sgRNAs read counts individually for the two condition. Histograms of sgRNA representation fold-changes between day 21 and day 0 should be used.

- did the author observed any enriched guides? i.e. targeting pseudo genes that upon inactivation, increased cellular fitness? They mention only their filtering strategies, accounting for depleted guides.

- it is not clear how/if the author checked that observing negatively selected parent genes was not due to off-target effects of the sgRNA targeting the corresponding pseudo-genes (due to their sequence similarity). The number of pseudo-genes and parent-genes hits is the same (69). Is it the case that for all the pseudo-gene hit also the corresponding parent gene is called as significantly impacting cellular fitness? combined with the larger effect magnitude of the parent genes with respect to their corresponding pseudo genes, this looks a bit suspicious to me and suggesting that it might actually be due sgRNA off targets effect. How the authors can exclude this? This control is performed only for the handful of guides put forward for further validation.

- Why the authors picked and decided to put forward only hits from screening the MCF7 cell line? What about those from the MDA-MB-231 cell line? shouldn't a consensus set of hits have been selected and put forward for further validation?

Reviewer 2

Are you able to assess all statistics in the manuscript, including the appropriateness of statistical tests used? Yes, and I have assessed the statistics in my report.

Comments to author:

The manuscript by Sun et al. investigates an intriguing area of genome biology--that of the function of pseudogenes. The authors design a custom CRISPR screen to systematically identify pseudogenes potentially implicated in cellular fitness. They identify 69 of 850 targeted

pseudogenes that have a possible cellular fitness effect. Among these, they investigate the MGAT4EP pseudogene more deeply, suggesting a role for its regulation of FOXA1 and FOXM1. Overall, the study is interesting. The experiments are well executed and done with careful consideration. In particular, the sgRNA library appropriately targets pseudogenes and parent genes, and the authors attempt to de-convolute bidirectional promoters. The authors also take effort to define appropriate 5' transcriptional start sites prior to sgRNA design. Overall, this is an intriguing study that tackles a challenging question in genome biology. The data are promising, and the concept is good. However, I have reservations about the execution of several key experiments that need to be addressed by the authors.

1. The negative controls for the CRISPR screen are unbalanced and unusual. There are reportedly 267 sgRNAs to AAVS1 and 83 non-targeting sgRNAs. This is not typical. Generally, there are hundreds of random genome-cutting sgRNAs to non-coding/non-transcribed loci, not just AAVS1. Likewise, typically there are hundreds of non-targeting sgRNAs. While I recognize that the authors cannot easily re-design their sgRNA library and re-perform the screens, I think that more negative control sgRNAs are required for their validation experiments (eg. Figure 3B-H). Specifically, these experiments should include two different genome-targeting sgRNA controls from different genomic regions (e.g. not two AAVS1-targeting sgRNAs).
2. The authors should investigate putative off-target effects of the sgRNAs more fully. The authors can analyze the number of genomic targeting sites by BLAT (as sgRNAs targeting many sites are confounded) as well as use CasOFFinder or another similar tool to predict off target effects of sgNRAs.
3. Rescue experiments should be performed for 4 pseudogenes validated by CRISPR assays in Figure 3C-F. This would presumably be overexpression of a sgRNA-resistant cDNA, which would obviate the loss-of-viability phenotype.
4. How do the authors know that the pseudogenes targeted here do not encode a protein or peptide? A subset of pseudogenes are thought to be translated by ribosomes (see Ji, eLife, 2015) and some have mass spectrometry peptides to support translation. The authors should ensure that MGAT4EP does not translate a protein via in vitro transcription/translation assays, and analysis of publicly-available mass spectrometry data.
5. Why is GAPDH mRNA not significantly enriched in the cytoplasm in Figure 4A-B? It seems that these experiments lack a cytoplasmic control.
6. For Figure 4D-E, the authors use GAPDH mRNA as a negative control for FOXA1 pull-down. GAPDH mRNA is not an appropriate negative control since it should be in the cytoplasm, and MGAT4EP should be located in the nucleus. The correct negative controls would be other nuclear RNAs that do not bind FOXA1. With the current data, it is possible that all nuclear RNAs bind FOXA1 non-specifically.
7. The concept of a sponge/ceRNA effect is controversial, with many publications on this topic but mostly of low-quality data. There is highly-quality evidence clearly suggesting that the

ceRNA hypothesis is largely invalid (Denzler, *Molecular Cell* 2014). Moreover, the data from original ceRNA paper (Poliseno, *Nature* 2010) has been difficult to reproduce (Kerwin, *Replication Study in eLife*, 2020). Some researchers suggest that any ceRNA effects are highly nuanced and highly context-specific, and do not occur with many miRNAs at all (Bosson, *Molecular Cell*, 2014). To address this, the authors should (1) ensure that MGAT4EP does not encode a protein responsible for FOXA1 regulation, and (2) generate a mutant control MGAT4EP that mutates the putative binding site between FOXA1 and MGAT4EP and include this for their pull-down assays in addition to the antisense RNA as a control.

Reviewer #1

General comments: “The authors present results from the first CRISPRi screen performed to date for interrogating human pseudogenes' function. They devised an effective bioinformatics pipeline to design a library of single guide RNA targeting pseudo genes, combining a CAGE analysis to identify transcription starting sites at a genome scale (including TSS of pseudogenes) with Sequence scan for CRISPR to identify possible target regions for sgRNAs. Secondly, they selected two breast cancer cell lines and focused only on pseudogenes targeted by at least 3 sgRNAs and expressed at the basal level in the two models (using public available RNA-seq data from the cancer cell line encyclopedia). The final library included also guides targeting pseudo-gene parent genes, plus positive and negative controls, i.e. core-fitness genes and non targeting guides. Using previously described experimental setting, the authors screened the two selected models with their library and were able to identify a number of pseudo-gene hits, that upon inactivation reduce cellular fitness. They selected the top hits and put them forward to further experimental validation, functional characterization and clinical relevance estimation. Among these, they extensively characterize the role of MGAT4EP as a cancer-testis unitary pseudogene promoting cellular fitness in breast cancer. Briefly, this is a great piece of work describing an unprecedented effort, setting standards for future studies employing CRISPRi screens to characterize pseudo genes and reporting results of potential impact. Particularly, the bioinformatics pipeline used to design the library is well thought, comprehensively described and the author included appropriate positives and negative controls. A number of points should be addressed and others clarified, in my opinion before proceeding with the publication of this manuscript in genome biology.”

Response: We have performed additional analyses to address the reviewer's concerns. Please find our responses to individual comments/points as follows.

Specific points:

Comment 1: “The authors should mention that the final library is specific to the two screened models (as they excluded pseudogenes not expressed in these two models).”

Response: Following the reviewer's suggestion, we have added a clarification in the Results section that as a proof-of-principle systematic study of human pseudogene function with CRISPRi screens, our focus is breast cancer and the library used in our CRISPRi screens is specific to the two breast cancer cell line models that represent distinct breast cancer subtypes: luminal A and triple negative/basal-like breast cancer in the revised manuscript.

Comment 2: “What is the final median number of sgRNA per pseudogenes? an histogram with number of sgRNA per pseudogenes should be included as a supplementary figure.”

Response: The final median number of sgRNAs per pseudogene is 6. Following the reviewer's suggestion, we have added a histogram showing the distribution of the sgRNAs per pseudogenes in the revised manuscript (Supplementary Fig. 1A).

Comment 3: “Among the parent genes included in the library is there any MCF7 or MDA-MB-231 specific essential gene? if so these could be used as further positive controls. The authors might use public available data from genome wide CRISPR-Cas9 recessive screens, available on project score (PMID: 33068406) and the cancer dependency map portal (<https://depmap.org/portal/>), to this aim.”

Response: Following the reviewer’s valuable suggestion, we have used the essential parent genes that were identified by previous CRISPR-Cas9 knockout screens in MCF7 cells as further positive controls (Supplementary Fig. 1D and E), based on the publicly available data curated in the project score database (PMID: 33068406). Indeed, the CRISPRi sgRNAs that target the essential parent genes identified by previous CRISPR-Cas9 knockout screen, showed a statistically significant larger fold change of decrease between day 21 and day 0, compared with the ones targeting all parent genes (*Mann-Whitney U test*, $p < 1.21 \times 10^{-4}$, Supplementary Fig. 1D). We only performed CRISPRi screens in the MCF7 cell line, although we originally planned to perform CRISPRi screens in both MCF7 and MDA-MB-231 cell lines to gain insight into the common and different hits from these two cell lines that represent different breast cancer subtypes. It was very unfortunate that, due to family reason during COVID-19 pandemic, the first author Dr. Ming Sun decided to end his postdoctoral training and went back to China to stay with his families during the pandemic. Therefore we were unable to start the CRISPRi screen in the MDA-MB-231 cell line, although we included the sgRNAs that target expressed pseudogenes in MCF7/MDA-MB-231 cell lines in our CRISPRi sgRNA library. We have added a clarification that the screens were only performed in luminal A breast cancer cells in the revised manuscript.

Comment 4: “The authors do not specify if the screens were performed in duplicated batches and how reproducibility across these was assessed.”

Response: Like many published CRISPR/Cas9 knockout screens or CRISPRi screens, our screens were conducted in triplicates in a single batch. We have added a clarification about this and added an analysis of the correlation in sgRNA abundance between different replicates to demonstrate data reproducibility (Supplementary Fig. 1B).

Comment 5: “The authors should discuss the rationale beyond the selection of the two models employed in their screen and also mention in title, abstract and introduction that their results are specific to breast cancer (they might actually be model specific, i.e. linked to genomic background of MCF7 and MDA-MB-231, this should be mentioned in the discussions).”

Response: Following the reviewer’s valuable suggestions, we have added a description of the rationale for the selection of the two models employed in our screen in the Results section. We have also mentioned in abstract and introduction that our proof-of-principle CRISPRi screens for pseudogenes are specific to breast cancer. Furthermore, we have added a discussion mentioning that the results we obtained might be model specific in the Discussion

section. Although the CRISPRi screens we performed are specific to breast cancer, the scope of our study is more general. First, we developed a genome-wide library of CRISPR interference (CRISPRi) sgRNAs that target human pseudogene and will greatly facilitate the study of human pseudogene function under diverse biological contexts. Second, we performed integrative analyses of multi-omic data from the Cancer Genome Atlas (TCGA) and revealed many unitary pseudogenes, whose expressions are significantly dysregulated and/or associated with overall/relapse-free survival of patients in diverse cancer types. Last but not least, our study is not only the first systematic functional interrogation of human pseudogene in breast cancer, but also the first one in any biological contexts. Therefore we keep the same title in the revised manuscript to reflect the general scope of our study that is beyond breast cancer.

Comment 6: “The legends of the supplementary figures are sloppy and including repetitions/errors, I would suggest a proper revision. Furthermore what is the difference between supplementary figure 1B and 1C ? Finally, effects on fitness are hardly readable if plotting sgRNAs read counts individually for the two conditions. Histograms of sgRNA representation fold-changes between day 21 and day 0 should be used.”

Response: We apologize for the inappropriate description of the supplementary figures, and we have updated the legends in the revised manuscript. As the original Figure S1C provides redundant information, we have removed it from the revised supplementary Figure 1. Following the reviewer’s suggestions, the histograms of both sgRNA-level and gene-level fold-changes between day 21 and day 0 have been added (Supplementary Fig. 1D and E).

Comment 7: “Did the author observed any enriched guides? i.e. targeting pseudo genes that upon inactivation, increased cellular fitness? They mention only their filtering strategies, accounting for depleted guides.”

Response: Yes, we observed some enriched guides, but did not identify any significant positively selected pseudogene hits with the filters of $p < 0.05$, $FDR < 0.25$ and $\log_2\text{Fold-Change} \geq \log_2(1.5)$. In the revised manuscript, we mentioned that we did not find any significant positively selected pseudogenes using the above-mentioned criteria. We also added the positive selection information provided by MAGeCK program for pseudogenes/parent genes to the Supplementary Table 3.

Comment 8: “It is not clear how/if the author checked that observing negatively selected parent genes was not due to off-target effects of the sgRNA targeting the corresponding pseudo-genes (due to their sequence similarity). The number of pseudo-genes and parent-genes hits is the same (69). Is it the case that for all the pseudo-gene hit also the corresponding parent gene is called as significantly impacting cellular fitness? combined with the larger effect magnitude of the parent genes with respect to their corresponding pseudo genes, this looks a bit suspicious to me and suggesting that it might actually be due to sgRNA off targets effect. How the authors can exclude this? This control is performed only for the handful of guides put forward for further validation.”

Response: Thanks for bringing up this important issue. We found that out of the 69 pseudogene and 69 parent gene hits, 15 pseudogenes and their corresponding parent genes (15) were both identified as hits. To investigate the potential off-targeting effect between the 15 pseudogene-parent-gene pairs, we used Cas-OFFinder, as suggested by reviewer #2, to predict the putative off-target sites of individual sgRNAs in the human genome. Because the off-target effect is much weaker when there are >1 nucleotides (nt) of mismatches (Nature Biomedical Engineering 2020; PMID: 31937939; Figure 5d) or there is any RNA/DNA bulge (Nature Biotechnology 2016, PMID: 26780180; Figure 5c, d and e) in the potential off-target sites, we focused on the predicted off-target sites with only 1-bp mismatch from a given sgRNA sequence. Among a total of 30 (15 pseudogene and 15 parent gene) hits, we found 6 of them have one and the only one significant negatively selected sgRNA ($p < 0.05$, $\log_2\text{Fold-Change} \leq -\log_2(1.5)$) that harbors a predicted off-target site within [-2kb,+1kb] from the TSS of its corresponding pseudogene/parent gene. After removing these putative functional off-targeting sgRNAs, four of the pseudogene/parent gene hits still have at least two significant negatively selected sgRNAs and two of them have one significant negatively selected sgRNA (Supplementary Table 3). These results indicate that the vast majority of the pseudogene/parent gene hits are not confounded by the off-targeting sgRNAs from their corresponding pseudogenes/parent genes. One caveat associated with our analysis of CRISPRi off-target effect is that it was based on the established knowledge about the off-target effect of CRISPR-Cas9 system that involves genome cutting, whereas CRISPRi is based on catalytically inactive Cas9 and does not involve genome cutting. Given our limited knowledge about CRISPRi-mediated off-target effect, the predictive power of such *in silico* analysis remains unclear. We have added a discussion about potential limitations of the *in silico* CRISPRi off-target effect analysis in the Discussion section.

Comment 9: “Why the authors picked and decided to put forward only hits from screening the MCF7 cell line? What about those from the MDA-MB-231 cell line? shouldn't a consensus set of hits have been selected and put forward for further validation?”

Response: We only performed the CRISPRi screen in the MCF7 cell line, although we originally planned to perform CRISPRi screens in both MCF7 and MDA-MB-231 cell lines to gain insight into the common and different hits from these two cell lines that represent different breast cancer subtypes. It was very unfortunate that, due to family reason during COVID-19 pandemic, the first author Dr. Ming Sun decided to end his postdoctoral training and went back to China to stay with his families during the pandemic. Therefore we were unable to start the CRISPRi screen in the MDA-MB-231 cell line, although we included the sgRNAs that target expressed pseudogenes in MCF7/MDA-MB-231 cell lines in our CRISPRi sgRNA library. We have added a clarification that the screens were only performed in luminal A breast cancer cells in the revised manuscript.

Reviewer #2

General comments: “The manuscript by Sun et al. investigates an intriguing area of genome

biology--that of the function of pseudogenes. The authors design a custom CRISPR screen to systematically identify pseudogenes potentially implicated in cellular fitness. They identify 69 of 850 targeted pseudogenes that have a possible cellular fitness effect. Among these, they investigate the MGAT4EP pseudogene more deeply, suggesting a role for its regulation of FOXA1 and FOXM1. Overall, the study is interesting. The experiments are well executed and done with careful consideration. In particular, the sgRNA library appropriately targets pseudogenes and parent genes, and the authors attempt to de-convolute bidirectional promoters. The authors also take effort to define appropriate 5' transcriptional start sites prior to sgRNA design. Overall, this is an intriguing study that tackles a challenging question in genome biology. The data are promising, and the concept is good. However, I have reservations about the execution of several key experiments that need to be addressed by the authors.”

Response: We have performed additional analyses and experiments to address the reviewer’s concerns. Please find our responses to individual comments/points as follows.

Specific points:

Comment 1: “The negative controls for the CRISPR screen are unbalanced and unusual. There are reportedly 267 sgRNAs to AAVS1 and 83 non-targeting sgRNAs. This is not typical. Generally, there are hundreds of random genome-cutting sgRNAs to non-coding/non-transcribed loci, not just AAVS1. Likewise, typically there are hundreds of non-targeting sgRNAs. While I recognize that the authors cannot easily re-design their sgRNA library and re-perform the screens, I think that more negative control sgRNAs are required for their validation experiments (eg. Figure 3B-H). Specifically, these experiments should include two different genome-targeting sgRNA controls from different genomic regions (e.g. not two AAVS1-targeting sgRNAs).”

Response: Thanks for raising this important issue. Unlike CRISPR-Cas9 knockout, CRISPRi mediated knockdown is based on the sgRNA guided dCas9-KRAB binding to the promoter region of target gene for transcription repression and does not involve genome cutting. Based on the published screens so far, a typical genome-wide CRISPR-Cas9 screen indeed includes more negative control sgRNAs than our screen. However, our screen is not a genome-wide screen and the library only includes ~9,400 sgRNAs that target pseudogene/parent gene, which is only ~1/10 of the number of gene-targeting sgRNAs (~100,000) in a genome-wide screen. Therefore less negative control sgRNAs were included in our screens compared with a genome-wide screen. For example, the popular human GeCKO v2 libraries (https://media.addgene.org/cms/files/GeCKOv2.0_library_amplification_protocol.pdf) contain ~120,000 sgRNAs that target 19,052 human genes and <2000 negative control sgRNAs designed not to target the genome (non-targeting sgRNAs). The ratio between the number of negative controls vs. the number of gene-targeting sgRNAs is actually higher in our library than GeCKO. v2 libraries.

The negative control sgRNA used in our validation experiments is a non-genome-targeting sgRNA (sg-NT). To control for the effect of different negative control sgRNAs, we have

performed additional validation experiments for the 4 pseudogenes with two new negative control sgRNAs, including one AAVS1-targeting sgRNA (sg-AAVS1) and one (sg-nAAVS1) that targets the region on chromosome 4 and is distant from AAVS1 on chromosome 19 (Supplementary Table 2). We selected the sg-AAVS1 because it showed insignificant fold-change in abundance in our CRISPRi screens. The sg-nAAVS1 was selected because it showed insignificant fold-changes in abundance across multiple cell lines in previous CRISPR-Cas9 knockout screens (PMID 28162770). Using these two genome-targeting negative controls, we successfully confirmed our previous findings. We have added these new results to the revised manuscript (Supplementary Fig. 2A-C).

Comment 2: “The authors should investigate putative off-target effects of the sgRNAs more fully. The authors can analyze the number of genomic targeting sites by BLAT (as sgRNAs targeting many sites are confounded) as well as use CasOFFinder or another similar tool to predict off target effects of sgNRAs.”

Response: Following the reviewer’s valuable suggestion, we further investigated the putative off-target effects of the sgRNAs on the results of our CRISPRi screen. Similar to BLAT, Cas-OFFinder performs a sequenced-based search for putative off-target sites, but provides a more flexible and powerful framework for executing this task than BLAT. Therefore we used Cas-OFFinder to identify the putative off-target sites of individual sgRNAs in the human genome. Because the off-target effect is much weaker when there are >1 nucleotides (nt) of mismatches (Nature Biomedical Engineering 2020; PMID: 31937939; Figure 5d) or there is any RNA/DNA bulge (Nature Biotechnology 2016, PMID: 26780180; Figure 5c, d and e) in the potential off-target sites, we focused on the predicted off-target sites with only 1-nt mismatch from a given sgRNA sequence. We found that most sgRNAs targeting pseudogene/parent gene were associated with no or very small number (≤ 1) of predicted off-target sites in the human genome (**Supplementary Fig. 1F**). Moreover, the number of predicted genomic off-target sites associated with off-targeting sgRNAs did not show significant difference between pseudogene/parent gene hits and non-hits (*Mann-Whitney U test*, $p \geq 0.18$, **Supplementary Fig. 1G**). Importantly, we found that the pseudogene/parent gene hits did not have a significantly larger proportion of off-targeting sgRNAs with a large number (≥ 10) of predicted off-target sites, compared with the other pseudogenes/parent genes (*Fisher’s exact test*, $p > 0.32$). Collectively, these results suggest that in overall, the potential off-targeting sgRNAs may have little impact on differentiating the screen hits from the other genes and thus the results of our CRISPRi screens. Aside from the global analysis of off-target effect, we further analyzed the potential off-target effects between identified pseudogene and parent gene hits and found the impact of putative off-target effects on the identified pseudogene/parent gene hits is little (see details in our response to Comment 8 from reviewer #1). One caveat associated with our analysis of CRISPRi off-target effect is that it was based on the established knowledge about the off-target effect of CRISPR-Cas9 system that involves genome cutting, whereas CRISPRi is based on catalytically inactive Cas9 and does not involve genome cutting. Given our limited knowledge about CRISPRi-mediated off-target effect, the predictive power of such *in silico* analysis remains unclear. We have added a discussion about potential limitations of the *in silico* CRISPRi

off-target effect analysis in the Discussion section.

Comment 3: “Rescue experiments should be performed for 4 pseudogenes validated by CRISPR assays in Figure 3C-F. This would presumably be overexpression of a sgRNA-resistant cDNA, which would obviate the loss-of-viability phenotype.”

Response: Following the reviewer’s suggestion, we performed the rescue experiments for the 4 pseudogenes validated by CRISPRi assays in Figure 3C-F, by overexpressing the corresponding cDNA upon CRISPRi-mediated knockdown. In these rescue experiments, no mutations was introduced into the cDNAs because CRISPRi mediated knockdown does not involve the cutting of the gene body region, but only affects the promoter region that is not part of the cDNA. Unlike CRISPR/Cas9 knockout, CRISPRi mediated knockdown is based on the sgRNA guided dCas9-KRAB binding to the promoter region of target gene for transcription repression. We found that overexpression of the corresponding pseudogene cDNA was able to rescue the loss-of-viability phenotype caused by the CRISPRi-mediated knockdown (Supplementary Fig. 2D). Furthermore, we found that cDNA overexpression rescued the loss-of-function phenotype in colony formation assay (Supplementary Fig. 2E). These results support that the observed loss-of-function phenotypes for these 4 pseudogenes are not due to off-target effect.

Comment 4: “How do the authors know that the pseudogenes targeted here do not encode a protein or peptide? A subset of pseudogenes are thought to be translated by ribosomes (see Ji, eLife, 2015) and some have mass spectrometry peptides to support translation. The authors should ensure that MGAT4EP does not translate a protein via in vitro transcription/translation assays, and analysis of publicly-available mass spectrometry data.”

Response: To rule out the possibility that MGAT4EP translates a protein, we first analyzed publically available and in-house (unpublished) ribosome profiling (ribo-seq) data in MCF7 cell line, and found no ribo-seq reads that support ribosome occupancy on MGAT4EP RNAs. Second, we predicted putative ORFs encoded by MGAT4EP using an ORF prediction module that solely relies on the sequence information and is implemented in our Ribo-TISH package (Nature Communications 2017 PMID: 29170441), and searched the publically available mass-spectrometry (MS) data in MCF7 and T47D cells for the MS/MS spectra that matched the protein sequences corresponding to these putative ORFs. We found no MS evidence of the candidate proteins encoded by these putative ORFs. Finally, we performed an *in vitro* translation assay and found no evidence of any protein products generated by MGAT4EP translation, with the appropriate positive and negative control (Supplementary Fig. 3E). Taken together, these results indicate that MGAT4EP does not translate a protein and functions as an ncRNA.

Comment 5: “Why is GAPDH mRNA not significantly enriched in the cytoplasm in Figure 4A-B? It seems that these experiments lack a cytoplasmic control.”

Response: To verify the quality of the nuclear and cytoplasmic fractionation experiments, we

collected both RNAs and proteins from the nuclear and cytoplasmic fractions. We used β -tubulin as the cytoplasmic protein control and histone H3 as the nuclear protein control. The western blot results of these two protein markers (Supplementary Fig. 3H) supports the good quality of our nuclear/cytoplasmic fractionation. We have also changed the way in which we present the results for the cytoplasm/nucleus enrichment of RNAs based on the new batch of data so that it is clearer that there is a significant enrichment of GAPDH mRNA in the cytoplasm (Figure 4A).

Comment 6: “For Figure 4D-E, the authors use GAPDH mRNA as a negative control for FOXA1 pull-down. GAPDH mRNA is not an appropriate negative control since it should be in the cytoplasm, and MGAT4EP should be located in the nucleus. The correct negative controls would be other nuclear RNAs that do not bind FOXA1. With the current data, it is possible that all nuclear RNAs bind FOXA1 non-specifically.”

Response: To rule out the possibility that all nuclear RNAs bind to FOXA1 non-specifically, we included two additional nuclear RNAs as negative controls: the U1 for the short nuclear RNAs (<200bps) and the MALAT1 for the long nuclear RNAs (>200bps). We found that FOXA1 did not bind to U1 or MALAT1 based on RIP experiments (see figure below). Because MGAT4EP is a long pseudogene RNA, the MALAT1 is a more relevant negative control and thus only the result for MALAT1 was included into the revised manuscript (Figure 4E).

Comment 7. “The concept of a sponge/ceRNA effect is controversial, with many publications on this topic but mostly of low-quality data. There is highly-quality evidence clearly suggesting that the ceRNA hypothesis is largely invalid (Denzler, Molecular Cell 2014). Moreover, the data from original ceRNA paper (Poliseno, Nature 2010) has been difficult to reproduce (Kerwin, Replication Study in eLife, 2020). Some researchers suggest that any ceRNA effects are highly nuanced and highly context-specific, and do not occur with many miRNAs at all (Bosson, Molecular Cell, 2014). To address this, the authors should (1) ensure that MGAT4EP does not encode a protein responsible for FOXA1 regulation, and (2) generate a mutant control MGAT4EP that mutates the putative binding site between FOXA1 and MGAT4EP and include this for their pull-down assays in addition to the antisense RNA as a control.”

Response: Thanks for the valuable comments and suggestions. For clarification, our data

suggested that MGAT4EP did not function via a sponge/ceRNA mechanism, but supported its nuclear localization/function because for sponge/ceRNA regulation to be effective, the RNA has to be mainly localized in the cytoplasm to be accessible to miRNAs. But we agreed with the reviewer that it would be important to point out the limitation about the ceRNA mechanism/hypothesis and we have added a description, including the relevant literatures, about the limitation/controversial aspect of the sponge/ceRNA regulation in the Introduction section. To ensure that MGAT4EP does not encode a protein responsible for FOXA1 regulation, we have performed an analysis of publically available ribosome profiling and mass-spectrometry data. We also performed an *in vitro* translation assay. Our results indicate that MGAT4EP does not translate a protein and functions as an ncRNA (see details in our response to #4 comment). To identify the regions in the MGAT4EP RNA that was required for its interaction with FOXA1, we generated four serial deletion mutants with the deletion of 1-700, 700-1400, 1400-2100 or 2100-2819 bps, respectively. The RNA pull-down of antisense, full-length and serial deletion mutants of MGAT4EP RNA followed by anti-FOXA1 western blotting showed that the deletion of 1400-2100 bps of MGAT4EP abolished its interaction with FOXA1 (Fig. 4D), suggesting that this region is critical for MGAT4EP-FOXA1 interaction. We have added these new results to the revised manuscript (Fig. 4D).

Second round of review

Reviewer 1

The authors have satisfactorily address all my previous points.

Reviewer 2

The revised manuscript improves upon the first submission. I have no further comments.